# Consideration of various aspects in a drift study of MH370 debris

**Oleksandr Nesterov**[1,2]

[1]Independent Consultant (Physial Oceanography & Coastal Engineering)
[2]Principal Coastal Engineer at DHI Water & Environment (S) Pty Ltd. in the present

*Correspondence to:* Oleksandr Nesterov (oleksandr.nesterov@gmail.com)

**Abstract.** On March 7, 2014, a Boeing 777-200ER aircraft operated by Malaysian Airlines on the route MH370 from Kuala Lumpur to Beijin abruptly ceased all communications and disappeared with 239 people aboard, leaving a mystery about its fate. The subsequent analysis of so-called satellite 'handshakes' supplemented by military radar tracking has suggested that the aircraft ended up in the southern Indian Ocean. Eventual recovery of a number of fragments washed ashore in several countries has confirmed its crash. A number of drift studies were undertaken to assist in locating the crash site, mostly focusing either on the spatial distribution of the washed ashore debris or efficacy of the aerial search operation. A recent biochemical analysis of the barnacles attached to the flaperon (the first fragment found in La Réunion) has indicated that their growth likely began in the water of 24°C, then the temperature dropped to 18°C, and then it rose up again to 25°C. An attempt was made in the present study to take into consideration all these aspects. The analysis was conducted by the means of numerical screening of 40 hypothetical locations of the crash site along the so-called 7[th] arc. Obtained results indicate the likelihood of the crash site to be located between 25.5°S and 30.5°S latitudes, with the segment from 28°S to 30°S being the most promising.

## 1 Introduction

On March 7, 2014, approximately 40 minutes after the takeoff, a Boeing-777 aircraft (registration 9M-MRO) operated by Malaysian Airlines (MAS) as MH370 on the route from Kuala Lumpur (Malaysia) to Beijing (China), abruptly ceased all data and voice communications, and disappeared with 239 people aboard, leaving investigators clueless about a possible cause. The subsequent analysis of so-called satellite 'handshakes' (Ashton et al., 2015) supplemented by the primary radar tracking has suggested that the plane turned back, crossed the Malay Peninsula along Malay-Thai border, then flew toward Nicobar Islands in the Strait of Malacca, where it finally turned into the Indian Ocean, as detailed by the Australian Transport Safety Bureau (ATSB, 2014a, b) and the Ministry of Transport Malaysia (MTM, 2017). Eventual recovery of a number of 9M-MRO fragments, which were washed ashore in several countries, has confirmed the crash of the aircraft in the Indian Ocean. A total of 27 suspected and confirmed fragments were found according to the report published by the Malaysian Safety Investigation Team for MH370 (MSIT, 2017) on March 27, 2017: 1 in La Réunion, 2 in Mauritius, 1 in Rodrigues, 6 in Mozambique, 5 in South Africa, 1 in Tanzania, and 11 in Madagascar, with the latest find in January 2017.

Shortly after the disappearance, the Australian Government, whose geographical responsibility for rescue and recovery covers a region of the Indian Ocean where the terminus of 9M-MRO path could have been located according to the satellite data (Ashton et al., 2015), established the Joint Agency Coordination Centre (JAAC, 2014) to assist in the search operation. The Australian Maritime Safety Authority (AMSA, 2014), JAAC and ATSB have conducted an extensive aerial search operation, which lasted from March 18 to April 27, 2014, but failed to locate any debris related to MH370. Although some objects were spotted from the air, subsequent attempts to recover them were unsuccessful. After expiration of the underwater locator beacon, a device emitting acoustic signal to facilitate underwater search, the Australian Government has commissioned an engineering company Fugro N.V., which specializes on marine and geotechnical surveys, to conduct a deep-water high-resolution sonar survey of the seabed. The search domain assigned to Fugro N.V. was a band-shaped area in the South Indian Ocean along the so-called 7[th] arc from approximately 36°S to 39.5°S latitudes, defined by the ATSB (2015), and then later refined by the Australian Defense Science and

Technology Group (Davey et al., 2016). The 7$^{th}$ arc is a geometric curve on the Earth surface, all points of which are equidistant from the satellite, through which the last 'handshake' was transmitted (Ashton et al., 2015). The actual 7$^{th}$ arc may slightly differ from the nominal arc due to the uncertainty in the altitude of the aircraft, as well as truncation and measurement errors in the data (e.g., ATSB, 2015; Davey et al., 2016). Despite such an unprecedented effort, the underwater search was unsuccessful, and it was finally called off in January 2017 (MTM, 2017).

A number of drift studies were undertaken since then to assist in locating the crash site. Prior to the discovery of the flaperon in La Réunion on July 29, 2015, the studies focused on the analysis of the efficacy of the aerial search, such as García-Garrido et al. (2015). Later, after the flaperon and other 9M-MRO fragments were found, the mainstream approach shifted to the analysis of the probabilities of debris to reach specific locations by known dates starting from various origins along the 7$^{th}$ arc: the series of numerical and experimental studies undertaken by Griffin et al. (2017) at the Commonwealth Scientific and Industrial Research Organisation (CSIRO); the screening of 25 hypothetical locations along the 7$^{th}$ arc by Pattiaratchi and Wijeratne (2016) at the University of Western Australia; the numerical modelling conducted by Ormondt and Baart (2015) at Deltares; the study conducted by Maximenko et al. (2015) at the International Pacific Research Center; the study by Jansen et al. (2016) at the Euro-Mediterranean Center on Climate Change; the analysis of drifter trajectories obtained from National Oceanic and Atmospheric Administration's (NOAA) Global Drifter Program (GDP) in relation to MH370 debris by Trinanes et al. (2016). An alternative approach was based on the reverse drift modelling: the studies conducted by the French Government meteorological agency Météo France (Daniel, 2016) and GEOMAR Helmholtz Centre for Ocean Research Kiel (Durgadoo and Biastoch, 2015). The latter, however, did not account for wind forcing, which presumably explains the large difference in its conclusions compared to other studies.

There is an ongoing disagreement between conclusions of these studies with regard to the most likely origin of the debris at the 7$^{th}$ arc: the latest report published by CSIRO (Griffin et al., 2017) recommends a new search area at around 35°S, backed by the earlier IPRC study (34°S to 37°S). In contrast, Pattiaratchi and Wijeratne (2016) have suggested the crash site to be more likely located between 28.3°S and 33.2°S, narrowing down earlier Jansen et al. (2016) estimates (between 28°and 35°S), being also consistent with Ormondt and Baart (2015). Assuming zero drift angle of the flaperon, Daniel (2016) favors the location north of 25°S, but south of 35°S if the drift angle was set to 18° to the left with regard to wind at the leeway factor of 3.29% - both the parameters experimentally established by the Direction générale de l'Armement (DGA). Griffin et al. (2017) disagrees with these parameters, but confirms observed non-zero drift angles between 0°to 30°, explaining this effect by the longwise asymmetry of the flaperon.

An important feature of the fragment found in La Réunion are barnacles attached to it. Although it was not possible to establish their age, according to De Deckker (2017), who conducted biochemical analysis of the barnacles at the Australian National University, the start of their growth was in the water of approximately 24°C, and then for some time the temperature ranged between 20°C and 18°C, and then it went up again to around 25°C. This additional information has not been previously considered in the drift studies.

Consequently, this study comprises the three major elements to assess the most likely origin of the debris:
(1). Efficacy of the aerial search campaign.
(2). Ambient water temperatures at the flaperon.
(3). Spatial distribution of washed ashore debris.

## 2 Modelling

A total of 40 hypothetical locations of the debris origin along the 7$^{th}$ arc were screened against the three selection criterion by the means of forward particle tracking technique. The deterministic forcing of each particle in an ensemble was governed by the balance of water and air drag forces, magnitudes of which were assumed to be proportional to the squared relative (with respect to the particle) speeds of the ambient water and air respectively. Surface current velocities were sourced from the Hybrid Coordinate Ocean Model (HYCOM), wind data was sourced from the Global Data Assimilation System (GDAS), as detailed in Section 2.2.2. The stochastic component was modelled using the random walk technique (e.g., Al Rabeh et al., 2000; DHI, 2009; Jansen et al., 2016). Numerical integration was performed in the geocentric Cartesian coordinate system.

All the particles in an ensemble were 'released' from a single starting point at the 7$^{th}$ arc. Four models with regard to the leeway and drift angle properties of particles were considered. After individual particle tracks were obtained, they were supplemented by respective sea surface temperatures (SST) extracted from the publicly available archives. A subsequent analysis was undertaken in a statistical manner to: (1) estimate the maximum ensemble coverages during the aerial search; (2) assess percentage of particles in ensembles, which could have reached La Réunion by July 29, 2015, and could have been subjected to the temperature variation matching the results of the barnacle analysis; and (3) compare spatial distribution of particles washed ashore with the locations, where MH370 debris were found.

### 2.1 Model description

#### 2.1.1 Assumptions

In the frame of this study it was assumed that a particle was subjected to the drag forces induced by water and wind. Parti-

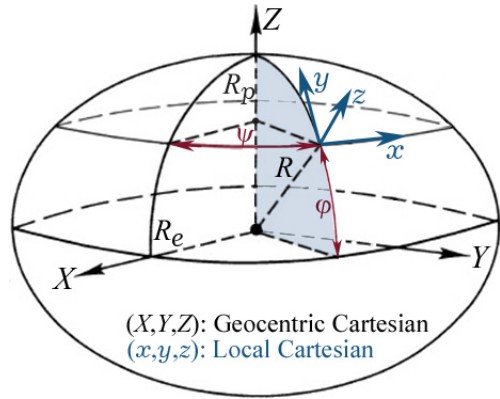

**Figure 1.** Local and geocentric Cartesian coordinate systems.

cles were assumed to be non-inertial. Impacts of the Coriolis force, Stokes' drift, waves, decay and sinking were neglected (e.g., Kraus, 1972; Al Rabeh et al., 2000). In the local coordinate system $(x, y)$, where the axes $x$ and $y$ correspond to the local west-to-east and south-to-north directions respectively (Figure 1), the location $(x_i, y_i)$ of the $i$-th particle was described by Langevin's equation (DHI, 2009):

$$\frac{dx_i}{dt} = \bar{u}_i + D(t, x_i, y_i)\zeta, \quad \frac{dy_i}{dt} = \bar{v}_i + D(t, x_i, y_i)\xi, \quad (1)$$

where $(\bar{u}_i, \bar{v}_i)$ is the average (deterministic) velocity of the $i$-th particle in the local coordinate system, $D$ is the turbulent diffusion term, and $\zeta, \xi$ are the random numbers, as detailed in Sections 2.1.3 through 2.1.5.

### 2.1.2 Coordinate systems

On the one hand, velocity of a particle and random walk are formulated in the local coordinate system, where the axis $z$ is normal to the Earth surface. On the other hand, the large study domain dictates the necessity to properly take into consideration the Earth curvature. To perform numerical integration of the governing equations in the geocentric Cartesian coordinate system $(X, Y, Z)$ depicted in Figure 1, where the Earth surface is approximated by WGS'84 ellipsoid with the polar and equatorial axes radii $R_p = 6356752$ m and $R_e = 6378137$ m respectively, transformation of coordinates and velocity vectors are required.

The Cartesian coordinates $X$, $Y$, and $Z$ of a point on the surface of the ellipsoid described by the longitude $\psi$ and geocentric latitude $\varphi$ can be formulated as:

$$X = R\cos\varphi\cos\psi, \quad Y = R\cos\varphi\sin\psi, \quad Z = R\sin\varphi, \quad (2)$$

where $R = \dfrac{R_e R_p}{\sqrt{(R_p \cos\varphi)^2 + (R_e \sin\varphi)^2}}$ is the distance

between this point and the center of the ellipsoid, as follows from the ellipse equation $\dfrac{(R\cos\varphi)^2}{R_e^2} + \dfrac{(R\sin\varphi)^2}{R_p^2} = 1$.

It should be noted that the backward transformation is required to extract and interpolate surface current and wind data at a location. Respective trigonometric transformations involve solving a $4^{th}$-degree polynomial equation.

The unit vectors $\overrightarrow{\mathbf{L}}$, $\overrightarrow{\mathbf{M}}$, and $\overrightarrow{\mathbf{N}}$, which define directions of the axes $x$, $y$, and $z$ of the local coordinate system (Figure 1) are introduced to obtain velocity components in the geocentric system. The outward unit vector $\overrightarrow{\mathbf{N}} = \{N_X, N_Y, N_Z,\}$ normal to the surface of the ellipsoid is formulated according to Korn and Korn (1968):

$$\overrightarrow{\mathbf{N}} = \frac{\left\{\dfrac{\partial X}{\partial \psi}, \dfrac{\partial Y}{\partial \psi}, \dfrac{\partial Z}{\partial \psi}\right\} \times \left\{\dfrac{\partial X}{\partial \varphi}, \dfrac{\partial Y}{\partial \varphi}, \dfrac{\partial Z}{\partial \varphi}\right\}}{\left|\left\{\dfrac{\partial X}{\partial \psi}, \dfrac{\partial Y}{\partial \psi}, \dfrac{\partial Z}{\partial \psi}\right\} \times \left\{\dfrac{\partial X}{\partial \varphi}, \dfrac{\partial Y}{\partial \varphi}, \dfrac{\partial Z}{\partial \varphi}\right\}\right|} \implies$$

$$N_X = \frac{1}{\sqrt{1+\mu^2}} \left(\cos\varphi - \mu\sin\varphi\right)\cos\psi,$$

$$N_Y = \frac{1}{\sqrt{1+\mu^2}} \left(\cos\varphi - \mu\sin\varphi\right)\sin\psi,$$

$$N_Z = \frac{1}{\sqrt{1+\mu^2}} \left(\sin\varphi + \mu\cos\varphi\right), \quad (3)$$

where $\mu = R^2 \dfrac{R_e^2 - R_p^2}{R_e^2 R_p^2} \cos\varphi \sin\varphi$.

The direction of the axis $x$ is defined by the unit vector $\overrightarrow{\mathbf{L}}$:

$$\overrightarrow{\mathbf{L}} = \{L_X, L_Y, L_Z\} = \{-\sin\psi, \cos\psi, 0\}. \quad (4)$$

Eq. (3) and Eq. (4) allow for expressing the unit vector $\overrightarrow{\mathbf{M}} = \{M_X, M_Y, M_Z\}$ collinear to the axis $y$ in the form of the vector product $\overrightarrow{\mathbf{M}} = \overrightarrow{\mathbf{N}} \times \overrightarrow{\mathbf{L}}$, so that its components are:

$$M_X = -\frac{1}{\sqrt{1+\mu^2}} \left(\sin\varphi + \mu\cos\varphi\right)\cos\psi,$$

$$M_Y = -\frac{1}{\sqrt{1+\mu^2}} \left(\sin\varphi + \mu\cos\varphi\right)\sin\psi,$$

$$M_Z = \frac{1}{\sqrt{1+\mu^2}} \left(\cos\varphi - \mu\sin\varphi\right). \quad (5)$$

Therefore, a velocity vector, the components of which are $\{\bar{u}, \bar{v}, 0\}$ in the local system, has the following components in the geocentric Cartesian system:

$$\overline{U} = -\bar{u}\sin\psi - \bar{v}\frac{1}{\sqrt{1+\mu^2}}\left(\sin\varphi + \mu\cos\varphi\right)\cos\psi,$$

$$\overline{V} = \bar{u}\cos\psi - \bar{v}\frac{1}{\sqrt{1+\mu^2}}\left(\sin\varphi + \mu\cos\varphi\right)\sin\psi,$$

$$\overline{W} = \bar{v}\frac{1}{\sqrt{1+\mu^2}}\left(\cos\varphi - \mu\sin\varphi\right). \quad (6)$$

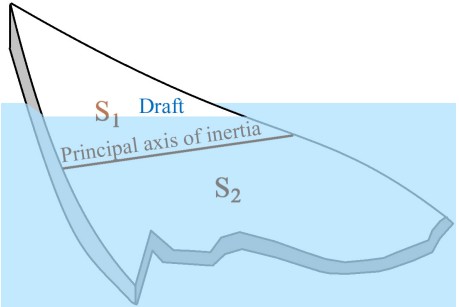

**Figure 2.** Schematic representation of a thin floating object.

Hence, Langevin's equation (1) can be formulated in the geocentric Cartesian system as follows:

$$
\begin{cases}
\dfrac{dX_i}{dt} = \overline{U}(\bar{u}_i, \bar{v}_i, \psi_i, \varphi_i) + D\left(L_X\zeta + M_X\xi\right), \\[2mm]
\dfrac{dY_i}{dt} = \overline{V}(\bar{u}_i, \bar{v}_i, \psi_i, \varphi_i) + D\left(L_Y\zeta + M_Y\xi\right), \\[2mm]
\dfrac{dZ_i}{dt} = \overline{W}(\bar{u}_i, \bar{v}_i, \psi_i, \varphi_i) + D\left(L_Z\zeta + M_Z\xi\right),
\end{cases}
\tag{7}
$$

where the relations between the longitude $\psi_i$ and geocentric latitude $\varphi_i$ of particle's location and its geocentric Cartesian coordinates $X_i, Y_i, Z_i$ is given by Eq. (2); the transformation of the velocity components is given by Eq. (6); and the local velocity components $\bar{u}_i$, and $\bar{v}_i$ are defined from the balance of the deterministic forces.

### 2.1.3   Deterministic terms

Mathematical description of the dynamics of a floating object is not a trivial problem due to the variety of processes and phenomenon in a near-surface layer, such as surface waves, Stokes drift, flow-object interaction, buoyancy, stratification, etc. (e.g., Kraus, 1972). Such an object is subjected to the dynamic pressure and shear stress forces due to the action of the water and air; its steady-state orientation depends on its buoyancy characteristics and the moments of forces around its principal axes of inertia. Denoting $\overrightarrow{\mathbf{u_i}} = (\bar{u}_i, \bar{v}_i)$ the deterministic velocity of the $i$-th particle representing a floating object in the local coordinate system, and neglecting the Coriolis force, its motion in the local horizontal plane is described by the equation:

$$
m_i\,\frac{d\overrightarrow{\mathbf{u_i}}}{dt} = \overrightarrow{\mathbf{F}}_{w,i} + \overrightarrow{\mathbf{F}}_{a,i},
\tag{8}
$$

where $m_i$ is the mass of the object, $\overrightarrow{\mathbf{F}}_{w,i}$ and $\overrightarrow{\mathbf{F}}_{a,i}$ are the average forces caused respectively by water and air flows around the object. The same formulation of these forces as applied in Daniel et al. (2002) and Breivik et al. (2011) to study drift of ship containers, was adopted in this study, although justification for thin nearly-horizontally floating objects (Figure 2) could be slightly different. According to

the theory of turbulent boundary layer (e.g., Kraus, 1972; Gandin et al., 1955), vertical velocity profiles of the water and air exhibit logarithmic dependence on the distance from the surface $U(z) = \dfrac{u_*}{\kappa}ln\dfrac{|z|}{z_0}$, where $u_*$ is the friction velocity, $\kappa = 0.41$ is von Kármán's constant, and $z_0$ is the roughness, where the surface is assumed to be non-moving at $z = 0$. The turbulent shear stresses $\tau_{\{w,a\}} = \rho_{\{w,a\}}u^2_{*\{w,a\}}$, where $\rho_{\{w,a\}}$ is the density of the water or air, remains constant through this layer. Hence, the shear stresses acting on the top and bottom surfaces of a thin floating object can be considered proportional to $\left|\overrightarrow{\mathbf{u}_w} - \overrightarrow{\mathbf{u}}\right|^2$ and $\left|\overrightarrow{\mathbf{u}_a} - \overrightarrow{\mathbf{u}}\right|^2$, where $\overrightarrow{\mathbf{u}_w} = \overrightarrow{\mathbf{u}_w}(x_i, y_i, t)$ and $\overrightarrow{\mathbf{u}_a} = \overrightarrow{\mathbf{u}_a}(x_i, y_i, t)$ are the current and wind velocities at the location of the object at certain reference height and depth (the typical reference height for wind is 10 m above the surface). The dynamic pressures can also be considered proportional to the squared relative velocities of water and air, but at the some other representative distances. Bearing in mind the logarithmic velocity profile, the latter would also be proportional to $\left|\overrightarrow{\mathbf{u}_w} - \overrightarrow{\mathbf{u}}\right|^2$ and $\left|\overrightarrow{\mathbf{u}_a} - \overrightarrow{\mathbf{u}}\right|^2$ respectively. Therefore, if the drag forces are collinear to the respective relative velocities of the water and air, the same formulation as in Daniel et al. (2002) would also be applicable for thin objects:

$$
\overrightarrow{\mathbf{F}}_{w,i} = \frac{1}{2}C_{Dw,i}\,S_{w,i}\,\rho_w\left|\overrightarrow{\mathbf{u}_w} - \overrightarrow{\mathbf{u}_i}\right|\left(\overrightarrow{\mathbf{u}_w} - \overrightarrow{\mathbf{u}_i}\right),
$$

$$
\overrightarrow{\mathbf{F}}_{a,i} = \frac{1}{2}C_{Da,i}\,S_{a,i}\,\rho_a\left|\overrightarrow{\mathbf{u}_a} - \overrightarrow{\mathbf{u}_i}\right|\left(\overrightarrow{\mathbf{u}_a} - \overrightarrow{\mathbf{u}_i}\right),
\tag{9}
$$

where $C_{Dw,i}$ and $S_{w,i}$ are the water drag coefficient and corresponding reference area of the submerged part of the object, $C_{Da,i}$ and $S_{a,i}$ are the air drag coefficient and corresponding reference area of the part exposed to the air, $\rho_w$ and $\rho_a$ are the water and air densities. Furthermore, Breivik et al. (2011) argued that wave drift forces on small objects (less than 30 m) decay rapidly, and they can be neglected compared to wind forces when the wave length is more than 6 times of the dimensions of a floating object.

It should be noted that composite materials are usually light-weight structures. For example, the recovered flaperon is of approximately 1.6 m x 2.4 m x 0.25 m size and 50 kg weight. Bearing in mind the typical values of the minimum drag coefficient $C_{Da}$ for airfoils are in the range 0.02 to 0.05 with respect to their surface areas, it is easy to show that the drag forces acting on such a fragment of the aircraft would normally be much larger compared to the inertial term in Eq. (8), and hence the latter can be approximated by the balance equation: $\overrightarrow{\mathbf{F}}_{w,i} = -\overrightarrow{\mathbf{F}}_{a,i}$. Using Eq. (9), the latter yields:

$$
\frac{1}{2}C_{Dw,i}S_{w,i}\rho_w\left|\overrightarrow{\mathbf{u}_i} - \overrightarrow{\mathbf{u}_w}(x_i, y_i, t)\right|\left(\overrightarrow{\mathbf{u}_i} - \overrightarrow{\mathbf{u}_w}(x_i, y_i, t)\right) =
$$

$$
-\frac{1}{2}C_{Da,i}S_{a,i}\rho_a\left|\overrightarrow{\mathbf{u}_i} - \overrightarrow{\mathbf{u}_a}(x_i, y_i, t)\right|\left(\overrightarrow{\mathbf{u}_i} - \overrightarrow{\mathbf{u}_a}(x_i, y_i, t)\right).
\tag{10}
$$

It is easy to see that the solution of Eq. (10) is:

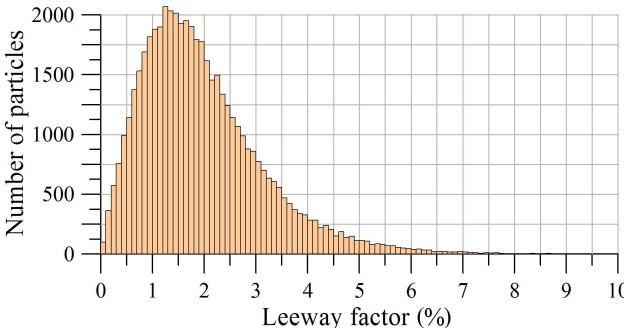

**Figure 3.** Assumed distribution of leeway factors across an ensemble (constant in time for an individual particle).

$$\overrightarrow{\mathbf{u}}_i = (1 - \alpha_i)\, \overrightarrow{\mathbf{u}}_w(x_i, y_i, t) + \alpha_i\, \overrightarrow{\mathbf{u}}_a(x_i, y_i, t), \tag{11}$$

where the scalar $\alpha_i$ (leeway factor) is:

$$\alpha_i = \frac{\sqrt{C_{Da,i}\, S_{a,i}\, \rho_a}}{\sqrt{C_{Da,i}\, S_{a,i}\, \rho_a} + \sqrt{C_{Dw,i}\, S_{w,i}\, \rho_w}}. \tag{12}$$

For a thin horizontally-floating object, which has equal areas of the surfaces exposed to the seawater and air ($S_a = S_w$), and which is characterized by equal drag coefficients $C_{Da} = C_{Dw}$, the leeway factor is $\alpha \approx 3.33\%$. This theoretical value is in a good agreement with the experimental data for the flaperon (3.29%) estimated by the DGA in the hydrodynamic engineering test facility center in Toulouse (Daniel, 2016). For a 70%-submerged cube, this formulation yields $\alpha = 2.2\%$ assuming the dominance of the dynamic pressure acting on the lateral faces, being also in a good agreement with the experimental factor of 2% (Breivik et al., 2011).

In two of the four considered models (Section 2.2), where the force induced by relative wind was not collinear to its direction, a modified formulation was used instead of Eq. (11):

$$\overrightarrow{\mathbf{u}}_i = (1 - \alpha_i)\, \overrightarrow{\mathbf{u}}_w + \alpha_i \begin{pmatrix} \cos\theta & -\sin\theta \\ \sin\theta & \cos\theta \end{pmatrix} \overrightarrow{\mathbf{u}}_a, \tag{13}$$

where $\theta$ is the drift angle, positive counter-clockwise.

### 2.1.4 Random leeway factor model

In contrast to earlier studies (e.g., Daniel, 2016; Pattiaratchi and Wijeratne, 2016; Griffin et al., 2017), an attempt was made in this work to take into consideration the variety of leeway factors, which describe random shapes and flotation characteristics of individual fragments generated by the crash. For this purpose, in one of the four considered types of ensembles (Section 2.2) particles were described by random leeway factors. It was assumed that these particles represented partially submerged thin objects of irregular shapes floating in slightly tilted orientations (Figure 2). For the sake of simplification, it was assumed that the drag coefficients for

the air and water are equal ($C_{Da,i} = C_{Ds,i}$). Then, according to Eq. (12), only the knowledge of the ratio of respective areas $k = S_{a,i}/S_{w,i}$ is required to estimate the leeway factor:

$$\alpha_i = \frac{\sqrt{\rho_a/\rho_w}\, \sqrt{k_i}}{1 + \sqrt{\rho_a/\rho_w}\, \sqrt{k_i}}. \tag{14}$$

In the frame of this study it was assumed that dimensions of individual objects are log-normally distributed. Furthermore, it was assumed that the principal axis of inertia of the object splits it into two parts, areas of which are also log-normally distributed, so that $k_i = \dfrac{\gamma_i\, S_{i,1}}{S_{i,2} + (1.0 - \gamma_i) S_{i,1}}$, where $\{\ln S_{i,1}, \ln S_{i,2}\} \in \mathcal{N}(\mu, \sigma^2)$, and $\gamma \in [0,1]$ is an independent random parameter to account for the draft of the object (0 - fully submerged; 1 - the center of gravity is at the water surface). Hence the logarithm of the ratio $\ln(S_1/S_2) = \ln S_1 - \ln S_2 \in \mathcal{N}(0, 2\sigma^2)$ is also normally distributed, with the mean of zero. Here the property was used that the sum of two independent normally distributed random variables is also normally distributed, with its mean being the sum of the means, and its variance being the sum of the two variances (Eisenberg and Sullivan, 2008). Hence, the ratio $S_1/S_2$ is log-normally distributed. The modelling was performed assuming $\ln(S_1/S_2) \in \mathcal{N}(0,1)$. The resulting distribution of the leeway factors of particles in an ensemble in this class of simulations (hereafter referred as the "random leeway" model) is depicted in Figure 3. A random leeway factor assigned to a particle in ensemble was constant in time.

### 2.1.5 Numerical realization and random walk

The whole integration interval was split into the time steps of $\Delta t = 15$ minutes. Similarly to DHI (2009) particle tracking model, integration of the system of equations (7) for each particle over the time step $\Delta t$ was comprised of the two stages, namely deterministic and stochastic:

$$\overrightarrow{\mathbf{X}}_i(t + \Delta t) = \overrightarrow{\mathbf{X}}_i(t) + \int_t^{t+\Delta t} \overrightarrow{\mathbf{U}}(\bar{u}_i, \bar{v}_i, \psi_i, \varphi_i)\, dt,$$

$$\overrightarrow{\mathbf{X}}_i(t + \Delta t) = \overrightarrow{\overrightarrow{\mathbf{X}}}_i(t + \Delta t) + \overrightarrow{\delta}_i(...),$$

where $\overrightarrow{\mathbf{X}}_i = \{X_i, Y_i, Z_i\}$ is the location of the $i$-th particle; $\overrightarrow{\overrightarrow{\mathbf{X}}}_i$ is the intermediate location prior to the superposition of the random displacement $\overrightarrow{\delta}_i(...)$; $\overrightarrow{\mathbf{U}}(\bar{u}_i, \bar{v}_i, \psi_i, \varphi_i) = \{\overline{U}(...), \overline{V}(...), \overline{W}(...)\}$ is the velocity of the particle. Unlike DHI (2009) model, which uses the 1-st order discretization method to integrate deterministic terms, the fifth- and sixth-order Runge-Kutta method was used in this work utilizing FORTRAN libraries to ensure particles remain on ellipsoid's surface with sufficient accuracy. During the integration, input current and wind data were bi-linearly interpolated in space, and linearly in time.

**Table 1.** Longitudes and latitudes of the origins preselected for screening

| No. | Lon.,°E | Lat.,°S | No. | Lon.,°E | Lat.,°S | No. | Lon.,°E | Lat.,°S | No. | Lon.,°E | Lat.,°S |
|---|---|---|---|---|---|---|---|---|---|---|---|
| **1** | 85.72 | 39.17 | **11** | 92.29 | 35.29 | **21** | 98.07 | 30.03 | **31** | 102.64 | 23.69 |
| **2** | 86.43 | 38.83 | **12** | 92.91 | 34.82 | **22** | 98.59 | 29.44 | **32** | 103.02 | 23.00 |
| **3** | 87.13 | 38.48 | **13** | 93.53 | 34.34 | **23** | 99.09 | 28.84 | **33** | 103.39 | 22.31 |
| **4** | 87.82 | 38.11 | **14** | 94.14 | 33.84 | **24** | 99.58 | 28.23 | **34** | 103.75 | 21.62 |
| **5** | 88.51 | 37.73 | **15** | 94.73 | 33.33 | **25** | 100.05 | 27.61 | **35** | 104.18 | 20.74 |
| **6** | 89.01 | 37.44 | **16** | 95.32 | 32.81 | **26** | 100.52 | 26.98 | **36** | 104.50 | 20.03 |
| **7** | 89.69 | 37.03 | **17** | 95.89 | 32.28 | **27** | 100.97 | 26.34 | **37** | 104.82 | 19.31 |
| **8** | 90.35 | 36.62 | **18** | 96.46 | 31.74 | **28** | 101.40 | 25.69 | **38** | 105.20 | 18.41 |
| **9** | 91.00 | 36.19 | **19** | 97.01 | 31.18 | **29** | 101.83 | 25.03 | **39** | 105.48 | 17.68 |
| **10** | 91.65 | 35.74 | **20** | 97.55 | 30.61 | **30** | 102.24 | 24.36 | **40** | 105.75 | 16.94 |

The vector $\overrightarrow{\delta_i}(...)$ corresponds to a numerical solution for the diffusivity term of the Langevin equation (1). It was treated as a random displacement in the $xy$-plane locally tangential to ellipsoid's surface. Its components $\delta_x$ and $\delta_y$ in the local coordinate system are the random values from the trimmed two-dimensional Gaussian distribution $\mathcal{N}_2$:

$$\delta_x = \sigma_L\,\zeta, \quad \delta_y = \sigma_L\,\xi, \tag{15}$$

where $\{\zeta, \xi\} \in \mathcal{N}_2(0,1)$ are the random numbers, and the standard deviation of the turbulent dispersion $\sigma_L = \sqrt{2\,D\,\Delta t}$ is assumed to be a function of the horizontal eddy diffusivity coefficient $D = D(t, \psi_i, \varphi_i)$. Such a relation between $D$ and $\sigma_L$ was first established by A. Einstein in 1905, who studied diffusion associated with Brownian motion, and since then it was adopted in a variety of random walk models (e.g., DHI, 2009; Jansen et al., 2016). In this study trimming was imposed to discard values $\delta_x$ and $\delta_y$ that resulted in displacements exceeding 10 km distance over the time step $\Delta t$. If this criteria was violated, the next pair of random values $\delta_x$ and $\delta_y$ was computed. The distance of 10 km was selected as the representative resolution of the ocean circulation model HYCOM, used as a source of the surface current velocities (see Section 2.2.2).

In contrast to Jansen et al. (2016), who applied the constant eddy diffusivity coefficient $D = 2$ m²/s, in this work $D$ was computed according to the well-known Smagorinsky (1963) parameterization, applied in various ocean circulation models (e.g., Blumberg and Mellor, 1987; DHI, 2009):

$$D = \kappa \Delta x \Delta y \sqrt{\left(\frac{\partial u}{\partial x}\right)^2 + \frac{1}{2}\left(\frac{\partial u}{\partial y} + \frac{\partial v}{\partial x}\right)^2 + \left(\frac{\partial v}{\partial y}\right)^2}, \tag{16}$$

where $\kappa = 0.1$ is the constant coefficient (a typical range is from 0.1 to 0.2 according to Blumberg and Mellor (1987)), $\Delta x$ and $\Delta y$ are the horizontal dimensions of the numerical grid cell, applied for the discretization of the velocity field. A discrete approximation of Eq. (16) was used to estimate the eddy diffusivity coefficient $D$, with subsequent bi-linear interpolation in space and linear interpolation in time at the actual locations of particles.

In the geocentric Cartesian coordinate system the random displacement translates into the 3D vector $\overrightarrow{\delta_i} = \{\delta X_i, \delta Y_i, \delta Z_i\}$, components of which are:

$$\delta X_i = L_X \delta_x + M_X \delta_y,$$
$$\delta Y_i = L_Y \delta_x + M_Y \delta_y,$$
$$\delta Z_i = L_Z \delta_x + M_Z \delta_y. \tag{17}$$

Such a displacement in the tangential plane causes a particle to move away from the surface of the ellipsoid. However, due to the imposed limitation on the distance, the elevation of the particle does not exceed 8 m after superposition of the random walk. Thus, particle elevations were forced to zeros after applying random walk procedure at each integration time step, while preserving longitudes and latitudes.

Box-Muller (1958) transform was used to obtain a pair of pseudo-random numbers $\{\zeta, \xi\} \in \mathcal{N}_2(0,1)$:

$$\zeta = \sqrt{-2\ln\tau}\,\cos(2\pi\omega), \quad \xi = \sqrt{-2\ln\tau}\,\cos(2\pi\omega), \tag{18}$$

where $\tau$ and $\omega$ are the two pseudo-random numbers from the interval $(0,1]$. To obtain $\tau$ and $\omega$, the use was made of the generator developed by Marsaglia and Zaman (1987), claimed to have the period of $2^{144}$.

Close to a shore, flow velocity was linearly interpolated between zero and velocity in the adjacent cell of the numerical grid of HYCOM. If a particle moved onshore as a result of wind action or random walk, all the subsequent forcing acting on such a particle was nullified, so that it remained at the location where it beached. No specific properties of the shores (e.g. type, slope), or contributing factors such as waves, tides, or storm surges, were considered.

It is worth noting that the transformations between the longitude, latitude and geocentric Cartesian coordinates were performed using extended precision accuracy (80 bits). Conducted tests have shown that the maximum errors in the elevation of a particle arising in a result of a single forward-backward conversion of the coordinates were of order 0.5 mm using the extended precision compared to 0.5 m using the double precision arithmetic.

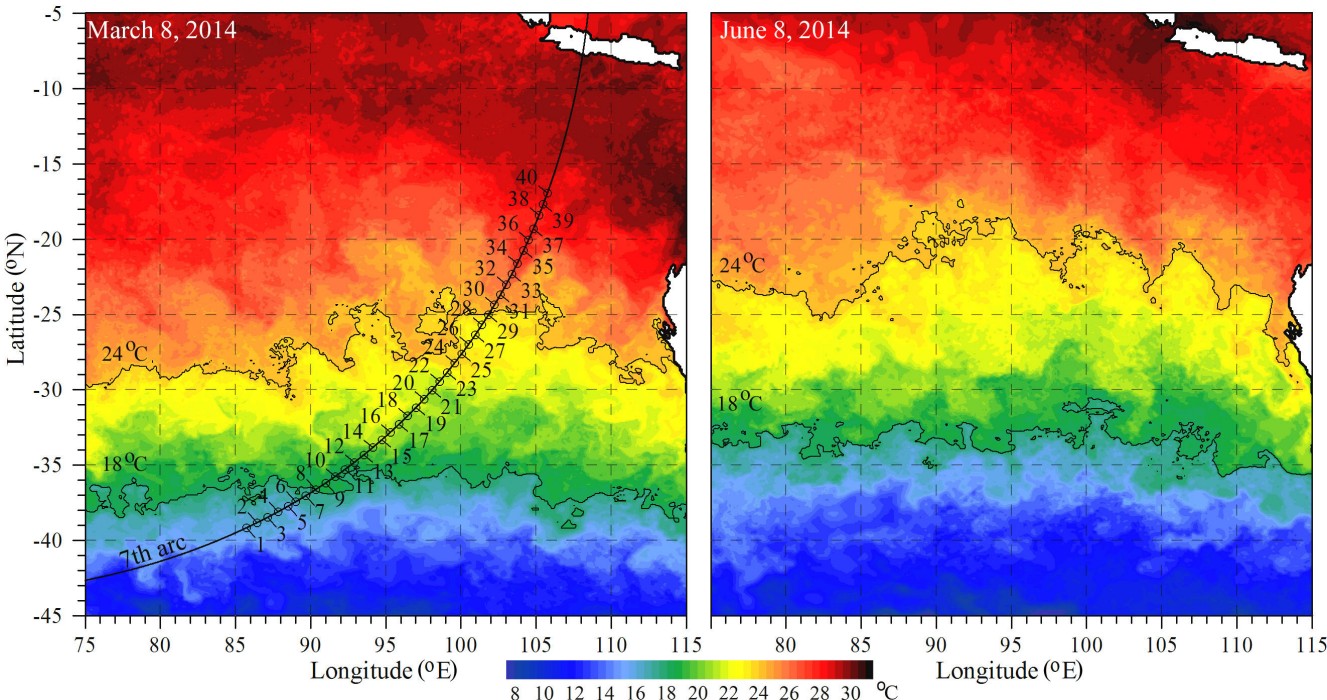

**Figure 4.** Locations of the selected hypothetical origins on the 7th arc, and snapshots of the sea surface temperatures.

## 2.2 Model setup

Four models with respect to the leeway factor and drift angle of particles in an ensemble were considered:

(1). Leeway factor of 3.29%; zero drift angle;

(2). Leeway factor of 3.29%; drift angle of 18° to the left;

(3). Leeway factor of 2.76%; drift angle of 32° to the left;

(4). Random leeway factor; zero drift angle.

All the ensembles released at various locations were identical with respect to the properties of particles in these ensembles. Each ensemble comprised 50,000 particles, same as in Pattiaratchi and Wijeratne (2016) study. The second and third models focused specifically on the flaperon path: respective settings corresponded to the flotation characteristics established by the DGA (Daniel, 2016). The fourth model aimed to achieve more realistic representation of the flotation characteristics of the debris generated by the crash by assigning random leeway factors (fixed in time) to the particles of an ensemble; all the 40 ensembles (Section 2.2.1) were identical in terms of the particles they were comprised of.

### 2.2.1 Screened locations

The study domain extended from 20°E to 140°E, 55°S to 15°N. Integration was performed from March 8, 2014 to December 31, 2016 inclusive. The locations of the 40 hypothetical debris origins are summarized in Table 1 and indicated in Figure 4. The coordinates of these locations were estimated from the burst time offset (ATSB, 2015) of the last 'handshake' 00:19, assuming that the aircraft was at the altitude of 10 km.

### 2.2.2 Model forcing and SST data

The following datasets were used in this study for the model forcing and temperature analysis:

- **Surface currents** were extracted from the Hybrid Co-ordinate Ocean Model, a data-assimilative isopycnal-sigma-pressure coordinate ocean circulation model (Chassignet et al., 2007). Spatial resolution: 0.08°x 0.08°; temporal resolution: daily. The HYCOM consortium is a multi-institutional effort sponsored by the National Ocean Partnership Program (https://hycom. org/); data are available at: ftp://ftp.hycom.org/datasets/global/GLBa0.08_rect/data/.

- **Wind velocities** were extracted from the Global Data Assimilation System, provided by the Air Resources Laboratory (ARL) of the U.S. National Oceanic and Atmospheric Administration (NOAA). Spatial resolution: 1°x1°; temporal resolution: 3 hours. Further details are available at http://www.ready.noaa.gov/gdas1.php; archived data in a proprietary format are available at: ftp://arlftp.arlhq.noaa.gov/pub/archives/gdas1.

- **SST** data were sourced from a Group for High Resolution Sea Surface Temperature (GHRSST) Level 4 MUR

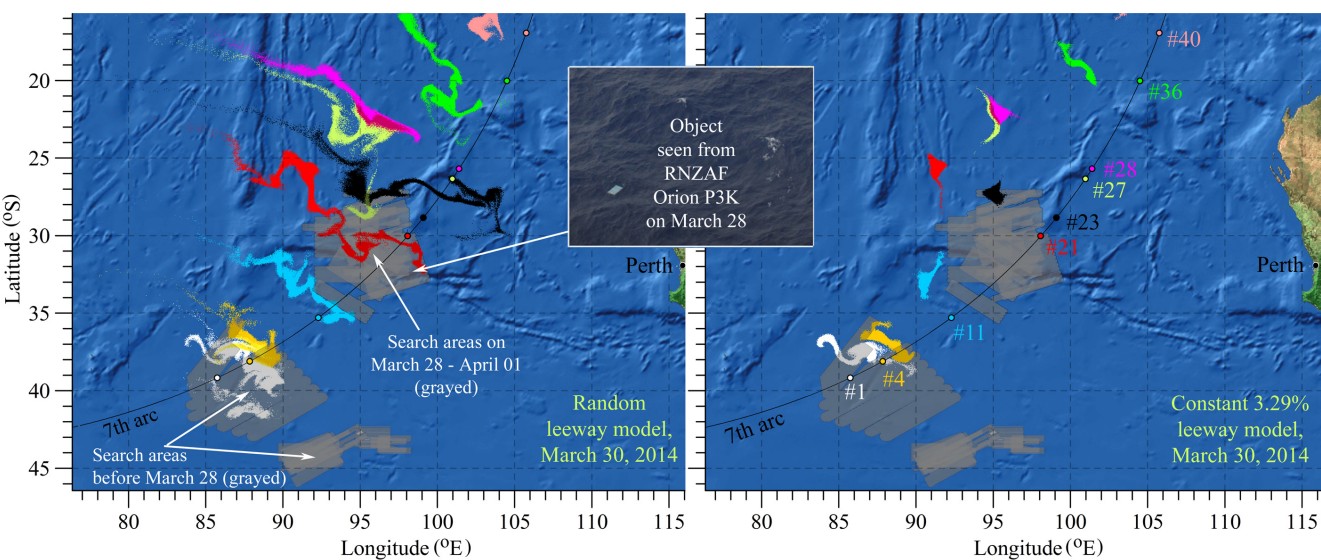

**Figure 5.** Snapshots of particle ensembles originating from several locations (indicated with corresponding colors of ensembles) on March 30 for two models, and the areas surveyed till April 2. Sources of the aerial search maps and photograph: JAAC (2014), AMSA (2014).

Global Foundation Sea Surface Temperature Analysis, a product developed by the Jet Propulsion Laboratory under NASA MEaSUREs program (JPL, 2015). Spatial resolution: 0.011°x 0.011°; temporal resolution: daily. Detailed information on this data is available at: https://mur.jpl.nasa.gov/ and https://podaac.jpl.nasa.gov/dataset/JPL-L4UHfnd-GLOB-MUR.

A direct comparison of the velocity components extracted and interpolated from HYCOM with those of the buoys available from the National Oceanic and Atmospheric Administration's Global Drifter Program (Eliot et al., 2016), which passed through the study domain from March 2014 till December 2016 (a total of 820,801 samples) have shown the RMS errors of 26.6 cm/s and 26.0 cm/s for the easterly and northerly components respectively. However, further analysis is required to understand contribution of wind to these errors (e.g., Griffin et al. (2017) suggests that the average leeway factor of the GDP buoys is around 1.2%), and, more importantly, how they can affect modelling accuracy, particularly with regard to whether they are stochastic or systematic.

## 3  Results

### 3.1  Efficacy of the aerial search

An extensive aerial search for MH370 debris, which lasted from March 18 to April 27, 2014, (e.g., JAAC, 2014; AMSA, 2014), failed to find any debris relevant to MH370. Although some suspected objects were observed on March 28 - 31 (AMSA, 2014), such as a rectangular object photographed from the specialized Royal New Zealand Airforce (RNAZF)

Orion P3K survey aircraft (Figure 5), subsequent attempts to recover them were unsuccessful.

Snapshots of the modelled particle locations in the ensembles originated from nine selected locations along the 7$^{th}$ arc on March 28 and April 5 are depicted in Figure 5 for the random and constant 3.29% leeway factor models. An Animation, which shows computed daily positions of these ensembles and search areas, is presented in the Supplement S1. As seen, the leeway factor had significant influence on the dispersion, which was notably more intense for the ensembles that comprised particles of various leeway factors. Furthermore, difference in the leeway factors could presumably explain the failed attempts to track suspected objects with the help of deployed buoys. It is worth noting that the aerial survey appears to be rather inefficient for the objects characterized by the leeway factor of 3.29% (such as the flaperon), which originated from the locations around 30°S or from the segment from 25.5°S to 28°S of the 7$^{th}$ arc.

To better understand reasons contributing to the aerial search failure, daily efficacies of the search were analyzed in terms of the percentages $E_{j,k}$ of particles $i_{j,k}$ of each $j$-th ensemble, which were in the search area $\Omega_k$ on the $k$-th day of the search:

$$E_{j,k} = \frac{1}{N}\left(\sum_{\Omega_k} i_{j,k}\right) \times 100\%, \qquad (19)$$

where $N = 50,000$ is the number of particles per ensemble. Respective areas $\Omega_k$ were obtained by digitizing maps published at AMSA (2014) and JAAC (2014) portals.

As the value of the maximum single-day coverage $\max_k E_{j,k}$ could be downplayed by other factors affecting detection capability, such as poor weather conditions, the five

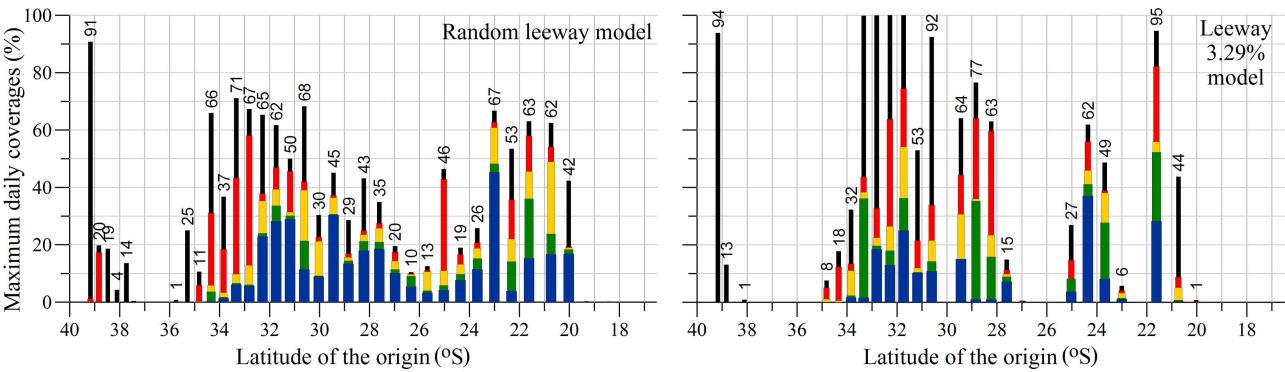

**Figure 6.** The five largest daily coverages during the aerial search (in the order of descent: black, red, yellow, green and blue bars).

**Table 2.** Maximum daily coverage (%), respective date of its occurrence (2014), and cumulative coverage (%) for each screened ensemble, the random leeway model.

| No. | Max. daily | Cumu-lative | Date | No. | Max. daily | Cumu-lative | Date | No. | Max. daily | Cumu-lative | Date | No. | Max. daily | Cumu-lative | Date |
|---|---|---|---|---|---|---|---|---|---|---|---|---|---|---|---|
| 1 | 90.7 | 91.9 | Mar 19 | 11 | 24.9 | 25.1 | Apr 01 | 21 | 30.4 | 114.1 | Mar 31 | 31 | 25.8 | 132.9 | Apr 04 |
| 2 | 19.8 | 37.2 | Mar 19 | 12 | 10.7 | 16.9 | Apr 01 | 22 | 45.1 | 279.0 | Mar 28 | 32 | 66.7 | 419.3 | Apr 04 |
| 3 | 18.6 | 18.6 | Mar 18 | 13 | 66.0 | 107.0 | Apr 01 | 23 | 28.6 | 116.1 | Apr 02 | 33 | 53.4 | 137.4 | Apr 05 |
| 4 | 4.3 | 4.3 | Mar 19 | 14 | 36.8 | 62.6 | Apr 01 | 24 | 43.1 | 192.3 | Mar 28 | 34 | 63.1 | 254.8 | Apr 05 |
| 5 | 13.6 | 13.6 | Mar 19 | 15 | 71.1 | 142.0 | Mar 31 | 25 | 34.9 | 199.1 | Mar 28 | 35 | 62.4 | 234.6 | Apr 05 |
| 6 | 0.4 | 0.4 | Mar 19 | 16 | 67.3 | 160.7 | Mar 31 | 26 | 19.5 | 136.2 | Mar 29 | 36 | 42.3 | 138.9 | Apr 03 |
| 7 | <0.1 | <0.1 | Mar 19 | 17 | 65.3 | 244.3 | Apr 01 | 27 | 10.5 | 68.8 | Apr 20 | 37 | 0.3 | 0.6 | Apr 05 |
| 8 | 0 | 0 | - | 18 | 61.7 | 295.1 | Mar 29 | 28 | 12.5 | 46.3 | Apr 18 | 38 | 0.2 | 0.6 | Apr 05 |
| 9 | <0.1 | <0.1 | Apr 01 | 19 | 50.0 | 256.8 | Mar 29 | 29 | 46.4 | 126.3 | Apr 13 | 39 | 0 | 0 | - |
| 10 | 0.8 | 0.8 | Apr 01 | 20 | 68.3 | 245.1 | Mar 29 | 30 | 19.0 | 100.8 | Apr 04 | 40 | 0 | 0 | - |

largest estimated daily coverages of particle ensembles are presented in Figure 6 for the random and constant 3.29% leeway factor models versus origin's latitudes along the 7th arc. The maximum single-day coverage, respective date of its occurrence, and cumulative coverage over the entire duration of the search campaign are summarized in Table 2 for each screened ensemble, random leeway factor model. The cumulative coverages were defined as the sum $\sum_k E_{j,k}$ of daily ensemble coverages over duration of the search campaign. It can be viewed as an additional selection criteria of a more likely origin when peak daily coverages are similar, such as in the case of ensembles No. 21 and 25 (Table 2).

As seen, if the crash site was located between 30.5°S and 34.5°S, or between 20°S and 25°S, debris would have been relatively well covered, which would consequently increase chances of their detection. The coverage of MH370 debris would have been notably lower if the crash site was located between 25.5°S and 27.5°S. In particular, for the origins No. 27 and 28, the maximum coverages of approximately 10% occurred more than 6 weeks after the crash, which is a rather low figure bearing in mind that decay and sinking processes were not taken into consideration. A relatively poor coverage is also noted for the origins No. 21 and 23.

Interestingly, a rectangular object spotted from RNZAF Orion P3K on March 28 was in the area, where some debris originating from the location No. 21 (98.07°E, 30.03°S) could be expected according to the random leeway model (Figure 5). Furthermore, the timing of the peak coverages (March 28 - 31; see Table 2) for the debris originating from this and neighboring locations is consistent with the dates when a number of floating objects were detected there.

## 3.2 SST along the path of the flaperon

One of the major goals of this study is to address a question whether the ambient water temperatures along modelled particle tracks could match the temperature variation derived from the biochemical analysis of the barnacles attached to the flaperon (De Deckker, 2017), and if yes, whether this information could help to further refine the search area. Therefore, those particles were of interest, which satisfied the two following conditions:

1. A particle must have arrived to the Sain-Andre beach, a place where the flaperon was found (assumed coordinates 55.67°E, 20.93°S), by July 29, 2015;

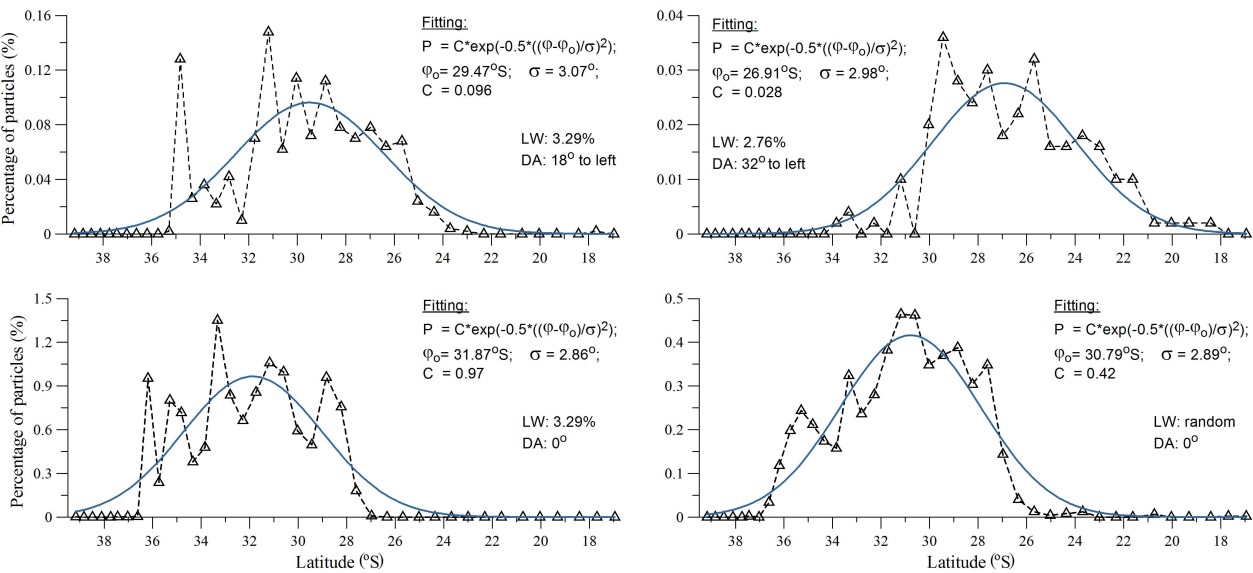

**Figure 7.** Percentages of particles satisfying distance and temperature criterion with the score S>0.01 (LW: leeway factor; DA: drift angle).

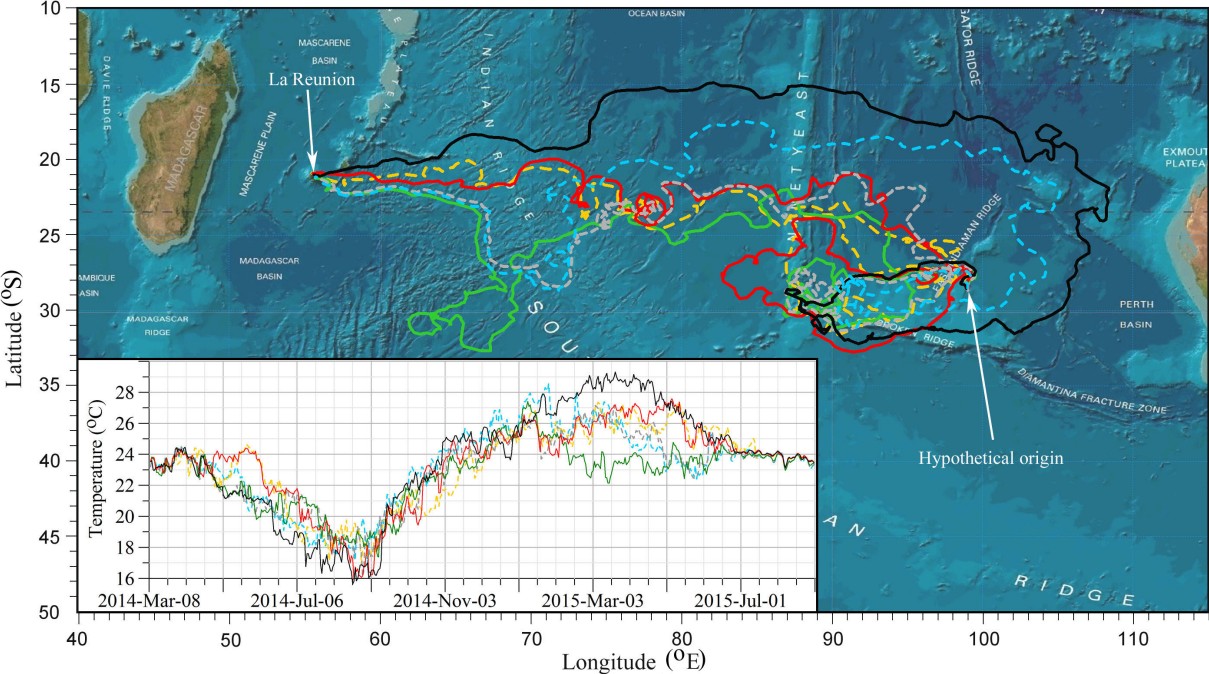

**Figure 8.** Sample tracks and temperatures for the 6 particles of the ensemble released at 99.09°E, 28.84°S (leeway 3.29%; drift angle 18°).

2. The ambient SST at particle's location should have first exceeded 24°C, then dropped down below 18°C, and then risen up again to 25°C or higher.

Due to the inherent uncertainty in the temperature estimations based on the barnacle analysis, a score-based function $S$ was introduced to identify those particles, which approximately satisfied the two aforementioned conditions:

$$S_i = S_{i,d}\ S_{i,\theta}. \tag{20}$$

Here the term $S_{i,d} = \exp(-0.5\ d_i^2/d_{ref}^2)$ is responsible for the first condition above; $d_i = d_i(\psi_i, \varphi_i)$ is the ground distance between the $i$-th particle location on July 29, 2015 and the Sain-Andre beach; $d_{ref} = 50$ km is the reference distance, which was chosen to be the approximate linear dimension of the Réunion Island. The term $S_{i,\theta}$ responsible for the second condition, was formulated in the following way:

$$S_{i,\theta} = \begin{cases} \dfrac{S_{i,\theta_1}+S_{i,\theta_2}+S_{i,\theta_3}}{3}, & \text{if } \exists\{t_1, t_2, t_3\} \in [T_s, T_e]: \\ & \max_{0 \leq t \leq t_1} \theta_i(t) \geq 23°\text{C}, \\ & \min_{t_1 \leq t \leq t_2} \theta_i(t) \leq 19°\text{C}, \quad (21) \\ & \max_{t_2 \leq t \leq T} \theta_i(t) \geq 24°\text{C}, \\ 0 & \text{otherwise}, \end{cases}$$

where $\theta_i(t)$ is the SST at the $i$-th particle location at the time $t$ ($T_s \leq t \leq T_e$, where $T_s$ and $T_e$ correspond to March 8, 2014 and July 29, 2015 respectively), and

$$S_{i,\theta_1} = \min(\max(\max_{0 \leq t \leq t_1} \theta_i(t) - 23, \ 0), \ 1),$$

$$S_{i,\theta_2} = \min(\max(19 - \min_{t_1 \leq t \leq t_2} \theta_i(t), \ 0), \ 1),$$

$$S_{i,\theta_3} = \min(\max(\max_{t_2 \leq t \leq T} \theta_i(t) - 24, \ 0), \ 1).$$

The purpose of such a formulation (20) is to relax the selection criteria and avoid discontinuities by assigning a positive score to a particle even if it did not arrive precisely to the Sain-Andre beach, or if it did not strictly satisfy the temperature condition (the tolerance allowing for a positive score was set to 1°C). As a result, the maximum score a particle could receive was one. If a particle arrived to La Réunion before July 29, it was still assigned a positive score. If a particle was located at the distance greater than 152 km from the Sain-Andre beach on July 29, it could not receive a score higher than 0.01. If the SST at particle's location never reached 23°C, or never dropped below 19°C after that, such a particle received zero score.

The percentages of particles in ensembles, which received scores S>0.01, vs. origin's latitudes along the 7th arc are shown in Figure 7 for all the four model setups, together with the normal distribution fitting. As seen, for the leeway factors and non-zero drift angles of the flaperon determined by the DGA (Daniel, 2016), the segment centered at 28.2±3°appears to be the most likely area, where the flaperon began its journey. A fraction of particles in the two other models (characterized by zero drift angle), which satisfied the two conditions, could reach a surprisingly high value of 1%, however, both the random leeway factor and constant 3.29% leeway factor models indicated the peak fitted probabilities at 30.8°S and 31.9°S respectively. The screened origins south of 36.5°S or north of 20°S are deemed to be unlikely starting locations of the flaperon, as none of the four models predicted notable percentage of particles meeting the two requirements above for the corresponding ensembles.

Examples of the first six particle tracks, which received the highest scores $S$, are depicted in Figure 8 for the scenario corresponding to the leeway factor of 3.29% and drift angle 18° to the left, starting from the origin No. 23 (99.09°E, 28.84°S). All these particles received the scores $S_i$>0.83, and they arrived to La Réunion between June 17 (black track) and

July 17 (yellow track), 2015. As seen, the two main reasons for the drop in the ambient water temperature from 23-25°C to as low as 16°C are:

1. Seasonal cooling of the water surface (see comparison of the SSTs on March 8 and June 8, 2014, in Figure 4);

2. Entrapment in counter-clockwise eddies, which could first carry the flaperon northwestward up to 22-25°S latitudes and then southward to 30-33°S (Figure 8).

### 3.3 Beached debris distribution

A total of 27 possibly relevant and confirmed fragments of 9M-MRO were recovered in La Réunion, Mauritius, Rodrigues, Madagascar, Mozambique, South Africa and Tanzania as of April 2017 according to the MSIT (2017); none was ever found in Australia, although a suspected object, the unopened towelette bearing MAS logotype, was discovered at Thirsty Point. Distribution of the found debris offers a useful insight into the possible location of the crash site (e.g., Jansen et al., 2016; Pattiaratchi and Wijeratne, 2016; Griffin et al., 2017), although no consensus was reached up to date with regard to the origin. Similarly to the aforementioned studies, an effort was made in this study to analyze spatial distribution of the washed ashore fragments.

A sample snapshot of the computed particle locations on December 31, 2016 is depicted in Figure 9 for the random leeway model, origin No. 23. Corresponding Animation S2a is included in the Supplement. Considerable beaching of particles of this ensemble was modelled in Africa, Madagascar, La Réunion and Mauritius, but negligible in Australia. The total fractions of particles landed by the ends of 2015 and 2016 are shown in the right top corner for all the screened origins, both the random and constant 3.29% leeway factor models. This result is, in part, due to the West Australian Current, which entrains large percentages of particles from the northern and southern origins (see Animations S3 and S4 in the Supplement), while most of particles from the middle of the screened segment of the 7th arc remain trapped in the Indian Ocean Gyre.

An interesting result can be obtained by comparing the modelled ratios of the fractions of particles washed ashore in several countries against the ratios of the number of fragments actually recovered in these countries. Such a comparison provides an indication of a number of fragments expected to be found by certain date assuming the same reporting factors. Figure 10 shows computed ratios for the random leeway model, using South Africa as the reference (a total of 5 objects were found there). As seen, more than 5 objects could be expected in Australia, at least 2 objects in Sri Lanka, 9 objects in Tanzania and 3 objects in Kenya, should the origin be south of 35°S. At least one fragment could be expected in Sri Lanka, and several in Kenya and Tanzania for the origins north of 23°S. The best matching segment of the 7th arc is located approximately between 26.5°S and 31°S. In particular,

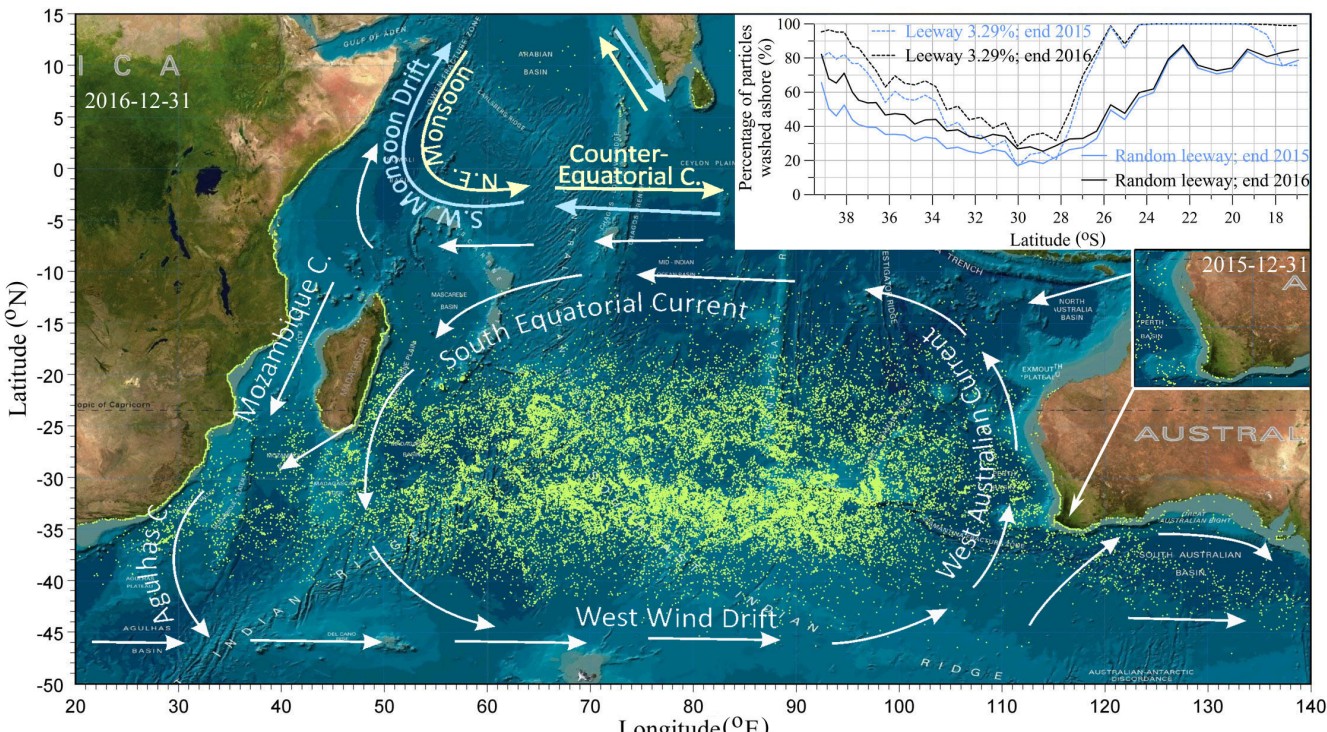

**Figure 9.** Snapshot of particle locations on Dec 31, 2016 (random leeway model; origin No. 23), and percentages of particles washed ashore.

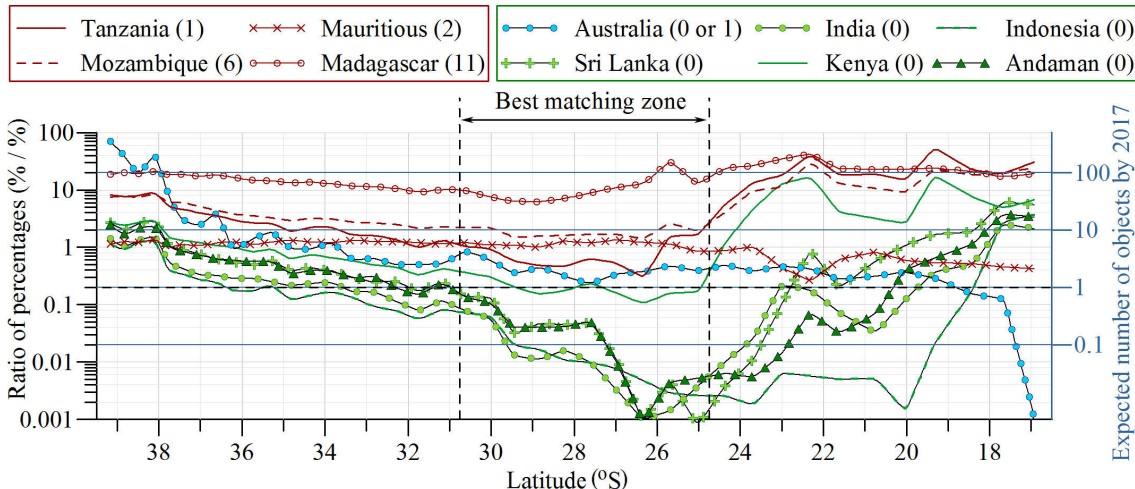

**Figure 10.** Expected number of objects to be found in several countries by Dec 31, 2016 vs. origin's latitudes.

the ratio of the percentages of particles beached in Mozambique to those in South Africa was in the range 1.4 to 1.7, being in a good agreement with the actual ratio of 6:5.

To estimate along-shore concentration of MH370 objects expected to be found by certain time, the two-dimensional Gaussian smoothing filter was first applied to obtain smoothed concentration of beached particles:

$$P(\psi,\varphi) = \frac{1}{2\pi d_{ref}^2} \sum_{i=1}^{M} \exp\left(-\frac{d_i^2(\psi,\varphi,\psi_i,\varphi_i)}{2\,d_{ref}^2}\right),$$

where $d_i$ is the ground distance between the $i$-th beached particle and the location of interest $(\psi,\varphi)$, $d_{ref}$ = 5 km is the size of the smoothing filter, and $M$ is the number of beached particles. To compute along-shore density distribution of objects expected to be found, this function was numerically integrated over a relatively narrow band-shaped areas $\Omega_i$ long-wise centered at the shorelines, divided by the respective lengths of the shoreline segments $\Delta s_i$, and prorated by the ratio of the number of fragments found in South Africa $N_{SA}$ = 5 to the number of particles landed in South Africa $M_{SA}$:

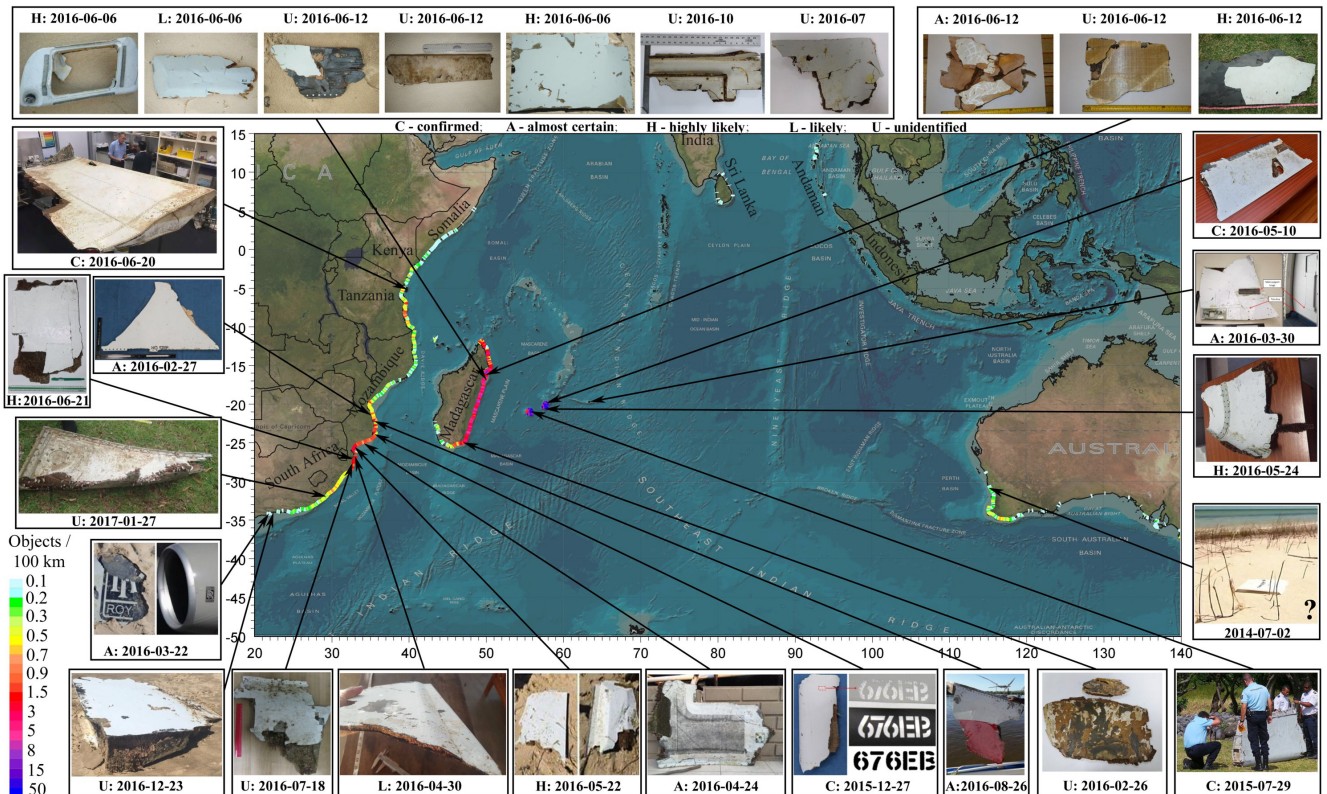

**Figure 11.** Expected along-shore concentration of beached MH370 fragments to be found by the end of 2016, origin No. 23.

**Table 3.** Sensitivity of the percentages of particles washed ashore by Dec 31, 2016 to the number of particles in ensembles (origin No. 23).

| Number of particles | Andaman Islands | Australia | India | Indonesia | Kenya | Madagascar | Mauritius | Mozambique | La Réunion | Somalia | South Africa | Sri Lanka | Tanzania | Overall beached | Escaped domain |
|---|---|---|---|---|---|---|---|---|---|---|---|---|---|---|---|
| **50,000** | 0.078 | 0.854 | 0.022 | 0.048 | 0.300 | 12.166 | 1.966 | 3.068 | 2.338 | 0.272 | 1.946 | 0.090 | 0.952 | 25.430 | 0.724 |
| **500,000** | 0.090 | 0.866 | 0.037 | 0.037 | 0.329 | 12.163 | 1.976 | 3.097 | 2.438 | 0.289 | 1.908 | 0.087 | 0.977 | 25.703 | 0.744 |

$$\frac{dN}{ds} \approx \frac{N_{SA}}{M_{SA}} \frac{1}{\Delta s_i} \iint\limits_{\Omega_i} P(\psi, \varphi) d\Omega.$$

An example of the so-estimated concentration of objects is shown in Figure 11 for the ensemble released from the origin No. 23 (random leeway model); concentrations for all the other studied origins are presented in Figure and Animation S5 in the Supplement. As seen, locations of the elevated concentrations are in a fairly good agreement with the locations, where the fragments were found, except the Rodrigues Island, which was not properly resolved by HYCOM. More fragments could be expected in Tanzania at 7°S and 8.2°S.

Screening of the origin No. 23 has revealed that the elevated concentration near Cape Leeuwin in Australia is due to the beaching, which mainly occurred in 2016. During 2014-

2015 notable arrival of particles from this origin to Australia took place only around the Windy Harbor and Thirsty Point, where the unopened MAS towelette was found on July 2 (assumed coordinates 115.3°E, 31°S). A detailed analysis has shown that 8 particles of the respective ensemble landed near Thirsty Point during July 8-11, and 10 more particles arrived before July 28, 2014. All the first 8 particles were characterized by relatively low leeway factors ranging from 0.1% to 0.4%. This suggests a systematic feature rather than a random occurrence. Systematic arrival of particles from the origins north of 27°S or south of 31°S was not predicted earlier than in the last week of July, as seen in Figure 12.

Figure 12 also shows the arrival times of particles to the Mossel Bay in South Africa, where the engine cowling fragment was found. Being the fourth found object, it had covered a long distance over a relatively short time interval.

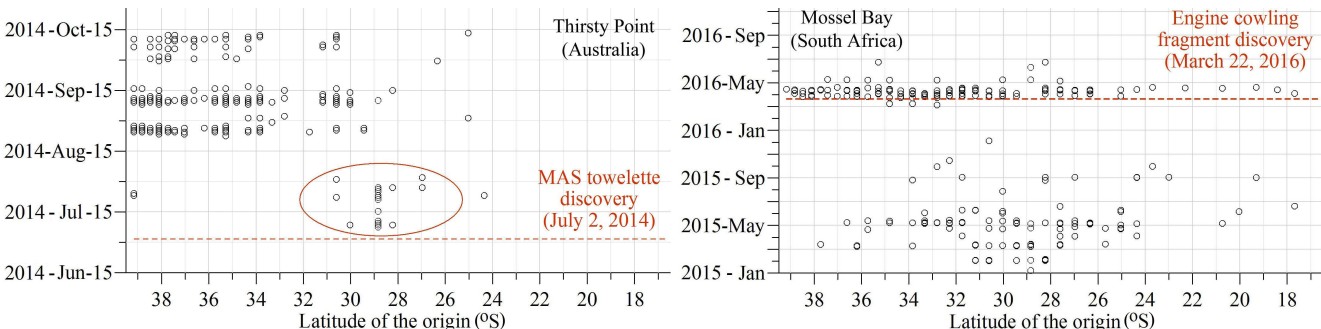

**Figure 12.** Arrival times of particles to the proximity of the Thirsty Point (Australia) during 2014 and Mossel Bay (South Africa).

The modelling has shown possibility of the sporadic arrival of high-windage particles to the Mossel Bay as early as in February 2015, but the systematic arrival was predicted only in March-May 2016, being consistent with the discovery date.

To understand sensitivity of the results presented in this section to the number of particles in ensembles, a numerical experiment was conducted using 500,000 particles for the origin No. 23, random-leeway factor model (see Animation S2b in the Supplement). The obtained results (Table 3) indicate that the use of 50,000 particles is sufficient to obtain fairly reliable statistics, and further increase of the number of particles would unlikely be beneficial.

## 4 Conclusions

The drift study of MH370 debris was conducted by the means of numerical modelling using forward particle tracking technique. A total of 40 hypothetical locations of the crash site along the 7th arc were screened. The three major aspects were considered: (1) efficacy of the aerial search; (2) ambient water temperatures along the path of the flaperon to La Réunion; (3) the spatial distribution of the washed ashore debris.

The governing equations were numerically integrated in the geocentric Cartesian coordinate system, where the Earth surface was approximated by the WGS'84 ellipsoid. Four models with respect to the leeway factors and drift angles were considered, including a proposed model of random distribution of the leeway factors of particles in an ensemble.

The obtained results indicate significance of the leeway factor in all the three aspects considered. In addition to the uncertainties in the model forcing, assumptions and simplification, judgment about the most likely location of the crash site depends on weights assigned to the aerial search, accuracy of the barnacle biochemical analysis, and probability of fragments not only to be washed ashore, but also recovered and reported. While it does not appear to be possible to confidently point out location of the crash site based on the drift study alone, a few observations can be made with regard to various segments of the 7th arc:

- **South of 36°S:** Considerable beaching where no fragments were found, particularly in Australia and Sri Lanka; incompatibility with the water temperatures suggested by the barnacle biochemical analysis.

- **34.5 to 36°S:** While corresponding areas were poorly surveyed during the aerial search, considerable beaching could be expected in several countries where no fragments were found, particularly in Australia.

- **30.5 to 34.5°S:** Excellent aerial coverage of the debris cloud originating from this segment makes the crash site unlikely to be located within it.

- **25.5 to 30.5°S:** Consistency with the barnacle temperature analysis; elevated concentration of beached particles, where the fragments of 9M-MRO were found; several 'gaps' in the aerial search; floating objects detected on March 28-31; possible consistency with the early arrival of the MAS towelette to Thirsty Point.

- **North of 25.5°S:** Inconsistency with the distribution of the washed ashore debris; incompatibility with the barnacle temperature analysis; good aerial coverage of the areas corresponding to the origins from 20°S to 25°S.

Summarizing all the above, the most likely area of the crash site appears to be between 25.5°S and 30.5°S, with the segment from 28°S to 30°S being the most promising. This area is consistent with the original definition of a high-priority search zone by the ATSB in June 2014.

*Competing interests.* This study was conducted solely at author's expense. The author declares that he has no conflict of interest.

*Acknowledgements.* Use was made of current data from HYCOM, provided by the National Ocean Partnership Program and the Office of Naval Research; the wind data was sourced from NOAA ARL GDAS; the SST data was obtained from NASA EOSDIS PODAAC at the JPL. The author is thankful to the two anonymous reviewers and topic editor, who helped to improve clarity and content of this paper.

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
