# Peer review of "Consideration of various aspects in a drift study of MH370 debris"

_Ocean Science, 2017_

## Referee Comment (RC1) · Anonymous Referee #1 · 17 Jan 2018

In this paper the author makes a very detailed study of the MH370 debris dispersion after its crash in the Indian Ocean. The study includes aspects that have not been previously considered, such as information about the temperature experienced by the flaperon, on its route to La Reunion where it was found, obtained from biochemical analysis. In my opinion the study provides very interesting conclusions and deserves publication in Ocean Science after clarifying some aspects that are unsettled in the current version of the manuscript.

1. The author provides an exhaustive bibliographic list, however I miss in this list the work by JA Trinanes et al. Journal of Operational Oceanography (2016).

2. Could the author provide references that explain how Eq. (8) is obtained? How

does it relate to the Maxey and Riley's equation of motion (Maxey and Riley, 1983) of a rigid spherical particle in a fluid ? Why Eq. (8) does not depend on the density of the floating object?

3. When integrating the equations of motion for each object there is a contribution of the ocean and wind velocity fields at each point. These velocity fields are subjected to uncertainties intrinsic to the models used to produce them. In this way HYCOM velocities differ from those supplied by other consortia producing global velocities, as for instance the COPERNICUS Marine Environment Monitoring Service. Could the author comment on how conclusions would be affected when using different velocity sources?

4. Simulations are performed considering ensembles of 50,000 particles. Movies such as S2, S3 or S4 confirm that this number of particles, in late stages of the simulations, leave numerous gaps and this could affect the conclusions obtained in these dates. Has the author obtained similar results considering higher number of particles in the ensembles as for instance 500,000 or 5,000,0000?. Additionally it would be useful if a date is included in each frame for movies S2, S3 and S4.

5. The random leeway factor used to produce figure 5, is a random distribution across the particle ensemble fixed in time or randomness varies in time?. Please clarify.

6. The paragraph explaining figures 5, and 6 and table 2 is unclear and thus the information contained in those figures and table is not understandable. The search areas marked in grey tone in Figure 5 were not considered during the whole period. For instance grey areas in the 1st column of figure 5 are early search areas, that were not considered in April. Why they appear in the second column? In the second column the grey areas appearing in the upper part were considered in later dates. Why are they sketched for April 5? Search areas change with time, therefore could the author explain how he defines cumulative coverages and why

they say something in this case? Could the author define how the magnitude represented in the vertical axis of figure 6 (Maximum daily coverage) is defined? What is the meaning of the color code in the bars of figure 6? What case, random or constant leeway, is explained in the paragraph from line 47 to 57? To what figure refers that paragraph? Table 2 is unclear: what kind of leeway uses?; what is the meaning of the dates there?; how magnitudes appearing in the upper row are defined?.

7. The autor states in lines 25-27 that: *If the SST at particle's location never reached 23ºC, or never dropped below 19ºC after that, such a particle received zero score.* However I do not see how this occurs by evaluating Eq. (18). For instance a particle which never reaches $23^oC$ but drops to $16^oC$ in the second period would score 0 in the first interval, 1 in the second time interval and 0 in the third one if it remains at $16^oC$. This according to (18) would score (0+1+0)/3 and this is not zero as stated by the author. Please clarify.

8. The information contained in figure S5 could be better interpreted if that figure is replaced by a movie made of the 40 frames displayed in the figure. It would be more intuitive if each frame included the initial point of the ensemble by marking it on the map instead of giving its geographical coordinates. The movie should include also the heading of the figure and the colorbar. The remaining information contained at the bottom of figure S5 could stay in a figure.

Please also note the supplement to this comment:
https://www.ocean-sci-discuss.net/os-2017-80/os-2017-80-RC1-supplement.pdf

---

## Referee Comment (RC2) · Anonymous Referee #2 · 20 Mar 2018

General comments =============== The disappearance of the Boeing 777-200ER of Malaysian Airlines in early 2014 is one of recent history's big mysteries. It is important for us as scientists and engineers to stay relevant to public interests and questions. I would like to see even more studies like the one presented here. I found the manuscript very interesting and relevant. The structure of the paper is well thought through and reads logically. I also found the figures very helpful in communicating both the study set-ups and finally the results. The use of biological information obtained from the barnacles on the recovered debris was probably my favourite part of the study especially when the modelling results could explain the temperature variations associated with the barnacle growth. In general, the four scenarios modelled, was also well constructed and adequately linked with the various debris drift scenarios.

[Figure]

Specific comments ================ The author did an adequate literature review. I would have like to see a bit more time spent on reviewing and/ or referencing other studies using a similar statistical approach. Or at least it would have been nice to get a clearer idea of how this drift study compares with other drift studies, not just studies related to MH370 debris. This is just a comment and not needed to make the manuscript acceptable. The coverage of other studies investigating the MH370 debris was well done and reads nicely. A non-technical person could read the introduction and get a clear idea of the context and relevance of the study. The three main aims listed on page 3 I also agree with. Refining the efficacy of search and rescue campaigns are crucial for future campaigns. As mentioned before, the cross correlation of the barnacle growth to surrounding water temperatures is my favourite part of the study and challenges us as scientist to think wider when it comes to answering scientific questions. I like the method of narrowing down the drift particles associated with debris discoveries on page 16. If the qualification 'window' for a particle was made to strict potential particle candidates, and thus traveling paths, might have been overlooked. As part of the conclusion it was reassuring to see that the study results agree with the high-priority search zone by the ATSB in June 2014.

Technical correction ================ • As I mentioned before I would have like to see more references. • In general, it is nice to read the introduction to a figure before the figure appears in the manuscript. The same goes for tables. • Equation 8 on page 6 was confusing due to unbalanced brackets. Please correct. Also, please reference literature used in this derivation or reference other studies using a similar technique or use literature to justify your approach. • Equation 10 on page 7 the drag coefficient CDs must surely be CDw?

---

## Author Comment (AC1) · 16 Apr 2018

Please see detailed responses in the attached pdf document.

Please also note the supplement to this comment:
https://www.ocean-sci-discuss.net/os-2017-80/os-2017-80-AC1-supplement.pdf

---

## Author Response (AR1)

**AUTHOR'S RESPONSE TO THE ANONYMOUS REFEREE'S #1 INTERACTIVE COMMENT.**

**General comment:** In this paper the author makes a very detailed study of the MH370 debris dispersion after its crash in the Indian Ocean. The study includes aspects that have not been previously considered, such as information about the temperature experienced by the flaperon, on its route to La Reunion where it was found, obtained from biochemical analysis. In my opinion the study provides very interesting conclusions and deserves publication in Ocean Science after clarifying some aspects that are unsettled in the current version of the manuscript.

**Author's response:** Thank you for your interest in my work, positive feedback and detailed comments. I hope that respective changes made to the manuscript will improve its clarity.
* * *
**Comment #1.** The author provides an exhaustive bibliographic list, however I miss in this list the work by JA Trinanes et al. Journal of Operational Oceanography (2016).

**Author's response:** This reference is indeed very relevant. I have found Trinanes et al. (2016) paper relatively recently, and that is why citation of it was missing in the original revision of the manuscript.

**Changed to manuscript:** Added into the new revision, along with citation "…*the analysis of drifter trajectories obtained from National Oceanic and Atmospheric Administration's (NOAA) Global Drifter Program (GDP) in relation to MH370 debris by Trinanes et al. (2016).*" in introduction.
* * *
**Comment #2.** Could the author provide references that explain how Eq. (8) is obtained? How does it relate to the Maxey and Riley's equation of motion (Maxey and Riley,1983) of a rigid spherical particle in a fluid ? Why Eq. (8) does not depend on the density of the floating object?

**Author's response:** Eq. (8) does not relate to Maxey and Riley (1983). In contrast to the latter, a relatively simple approach is considered in this work, same as in Daniel et al., (2002), and Breivik et al. (2011) – these references are added into the manuscript. In addition, floating aircraft's debris would rather be better represented by thin plates than spherical particles, as seen in a few images below and in Figure 11 of the manuscript (note that none of the so far recovered fragments are of spherical shape).

[Figure]

[Figure]

**Figure RC1. 1** The flaperon (on the left) and floating "No Step" (right horizontal stabilizer panel) fragment (on the right). The floatation of "No Step" fragment was apparently recorded by Blaine Alan Gibson, who discovered it in Mozambique, and posted by journalist Jeff Wise at https://www.youtube.com/watch?v=a7XcoRb8QIY.

Assuming that particle represents an object of the weight $m$, and neglecting Coriolis force, its motion in the local horizontal plane is described by equation:

$$m\frac{d\vec{u}}{dt} = \vec{F}_w + \vec{F}_a,$$

where $\vec{u}$ is the velocity (2D vector), $t$ is the time, $\vec{F}_a$ is the drag force induced by the air flow around the object, $\vec{F}_w$ is the drag force caused by water flow around the object. Here it is assumed that the weight of the object is balanced by its buoyancy. In general, the problem to calculate forces $\vec{F}_a$ and $\vec{F}_w$ is very complex, in part due to the motion and rotation of the object; see for example Chiang C. Mei "The Applied Dynamics of Ocean Surface Waves" (1989). Hence a simple quadratic dependence of the magnitudes of these forces on the relative water and air velocities was assumed, as in Daniel et al. (2002):

$$\left|\vec{F}_w\right| = \frac{1}{2}C_{Dw}\,S_w\rho_w\,(\vec{u_w} - \vec{u})^2, \quad \left|\vec{F}_a\right| = \frac{1}{2}C_{Da}\,S_a\rho_a\,(\vec{u_a} - \vec{u})^2,$$

where $\vec{u_w}$ and $\vec{u_a}$ are the water and air velocities, $\rho_w$ and $\rho_a$ are the water and air densities, $C_{Dw}$ and $C_{Da}$ are the drag coefficients for the parts of the objects exposed to the water and air respectively, $S_w$ and $S_a$ are the areas exposed to the water and air respectively.

The Boeing-777 flaperon is a composite material structure of approximately 1.6 m x 2.4 m size and 50 kg weight (Continental Airlines Inc. Boeing-777 Training Manual, Rev. 2). This suggests the bulk density of the flaperon-kind object to be approximately 100 kg/m³. It is useful to recall here that the drag coefficient of a typical airfoil is $C_{Da}$ = 0.045 in relevance to its surface area $S$; hence, assuming that $C_{Da}$ = $C_{Dw}$ , the drag forces induced by water and air flows around the flaperon with the relative speeds $U_{wr} = |\vec{u_w} - \vec{u}|$ and $U_{ar} = |\vec{u_a} - \vec{u}|$ would respectively be:

$$\left|\vec{F}_w\right| \approx 0.0225\,\rho_w\,S\,U_{wr}^2 \gg m\frac{U_{wr}}{T_H}, \quad \left|\vec{F}_a\right| \approx 0.0225\,\rho_a\,S\,U_{ar}^2 \gg m\frac{U_{ar}}{T_G}.$$

where $T_H$ = 86400 s and $T_G$ = 10800 s are the time resolutions of HYCOM and GDAS respectively, if $U_{wr} \gg 0.65$ mm/s and $U_{ar} \gg 44$ mm/s. This indicates that the magnitude of the inertial force is relatively small, and thus it can be neglected. This leads to the simple balance equation:

$$\vec{F}_a + \vec{F}_w = 0.$$

The later was presented by Eq. (8) in the manuscript. Note that in the revised manuscript it became Eq. (10).

It is worth of noting that the wind from the meteorological model GDAS is given at the standard reference height of 10 m (typical height of meteorological model's output) above the water surface, which is significantly higher than the elevation of floating objects in question, such as the flaperon or "No Step" fragment. The wind-induced drag can be split into the two components, namely the shear stress and dynamic pressure:

$$\vec{F}_a = \vec{F}_{as} + \vec{F}_{ad}.$$

Let's consider the atmospheric turbulent boundary layer, where the wind speed is described by the logarithmic profile:

$$U_a(z) = \frac{u_*}{k}\ln\frac{z}{z_0},$$

where $u_*$ is the constant dynamic velocity, $k$ = 0.41 is von Karman's constant, and $z_0$ is the roughness. The turbulent shear stress:

$$\tau = \rho_a u_*^2 \,,$$

is constant through the turbulent boundary layer. Hence

$$\left|\vec{F}_{as}\right| \sim \tau \;\Rightarrow\; \left|\vec{F}_{as}\right| \sim u_*^2 \;\Rightarrow\; \left|\vec{F}_{as}\right| \sim (U|_{z=10m})^2.$$

The other term, the dynamic pressure, can be assumed to be proportional to the squared relative speed at certain height $z_0 \ll z_1 \ll 10$ m:

$$\left|\vec{F}_{ad}\right| \sim \left(U|_{z=z_1}\right)^2 = \left(\frac{u_*}{k} ln \frac{z_1}{z_0}\right)^2 \;\Rightarrow\; \left|\vec{F}_{ad}\right| \sim (U|_{z=10m})^2.$$

Hence both the shear stress and dynamic pressure are proportional to the relative wind speed at 10 m height.

The same considerations are applicable with respect to the water in the near-surface layer; see, for example, Burchard H. and Pettersen O. "Models of turbulence in the marine environment — a comparative study of two-equation turbulence models", Journal of Marine Systems, 21, pp. 29-53, 1999. Therefore, the drag force induced by water can be considered proportional to the squared relative velocity of the water at certain reference depth below the surface, and the drag force induced by the air can be considered proportional to the squared relative velocity of the air at certain reference height above the surface.

**Changed to manuscript:** Section 2.1.3 was revamped. In particular two references and justifications applicable to thin floating objects were added.
* * *
**Comment #3.** When integrating the equations of motion for each object there is a contribution of the ocean and wind velocity fields at each point. These velocity fields are subjected to uncertainties intrinsic to the models used to produce them. In this way HYCOM velocities differ from those supplied by other consortia producing global velocities, as for instance the COPERNICUS Marine Environment Monitoring Service. Could the author comment on how conclusions would be affected when using different velocity sources?

**Author's response:** I have not tried to repeat modelling using other sources of current and wind data. I think it would require considerable efforts, while it would be more beneficial to compare input forcing with measured data. To analyze accuracy of HYCOM in the study domain, I have extracted velocity components of drifting buoys form NOAA's GDP database (Elipot et al., 2016, "A global surface drifter dataset at hourly resolution", J. Geophys. Res. Oceans, 121; www.aoml.noaa.gov; ftp://ftp.aoml.noaa.gov/pub/phod/buoydata), that passed through the study domain from March 2014 till December 2016 inclusive. Figure RC1.2 below depicts comparison of the components of velocities obtained from the buoys and extracted/interpolated from HYCOM, along with the distribution of errors per component. The total number of samples in this comparison is 820,801. The RMS errors were found to be 26.6 cm/s and 26.0 cm/s for U- and V- components respectively.

[Figure]

**Figure RC1. 2** Comparison of the U-component (west to east) and V-component (south to north) of the velocity obtained from NOAA GDP buoys and HYCOM.

However, it is necessary to note that:

(a) HYCOM data is daily product, while NOAA GDP data used in this comparison is hourly data.
(b) GDP buoys are likely to be affected by winds.

In general, I think random fluctuations, both in time and space, with the means of zeros can and should be treated with the random walk, subject to whether turbulent diffusion coefficient is adequately represented. This would require a more detailed analysis.

The effect of systematic errors, however, can be more pronounced. I have not figured yet an approach to deal with spatial and temporal variabilities to make a fair comparison between HYCOM with NOAA GDP data, but so far I have not observed systematic deviations.

Finally, it is not clear how the errors in the current and wind forcing can affect findings of this study, and what sensitivity tests should be conducted in this regards. I think spatial distribution of washed ashore debris can be somewhat affected due to the prolonged drift durations. For example, a systematic error as small as 1 mm/s in the current speed would result in 63 km distance error over 2 years. The conclusions derived from the aerial search survey could also be affected, but due to the small-scale eddies that could be present at the crash site, and due to wind, which has significant impact

on the initial dispersion due to the variety of leeway factors characterizing fragments. I believe the results of the temperature analysis would be less sensitive because of the following reasons:

- The drift characteristics of the recovered falperon are established experimentally;
- To experience the drop in the ambient water temperatures found by De Deckker based on the biochemical analysis of the barnacles, the flaperon would have to complete a relatively large circle (see Fig. 8). A travel, say, of 500 km south and then 500 km north during 2 months would imply the average southward and then northward speeds of approximately 20 cm/s (each over a month), which is comparable to the RMS error in HYCOM data (directly compared to NOAA GDP data). This would correspond to the systematic errors in HYCOM, which in general appear to be smaller.

**Changes to manuscript:** A paragraph is added into Section 2.2.2 "*A direct comparison of the velocity components extracted and interpolated from HYCOM with those of the buoys available from the National Oceanic and Atmospheric Administration's Global Drifter Program (Eliot et al., 2016), which passed through the study domain from March 2014 till December 2016 (a total of 820,801 samples) have shown the RMS errors of 26.6 cm/s and 26.0 cm/s for the easterly and northerly components respectively. However, further analysis is required to understand how these errors can affect modelling accuracy, particularly with regard to whether they are stochastic or systematic.*", as long as the reference to Eliot et al. (2016).
* * *
Comment #4. Simulations are performed considering ensembles of 50,000 particles. Movies such as S2, S3 or S4 confirm that this number of particles, in late stages of the simulations, leave numerous gaps and this could affect the conclusions obtained in these dates. Has the author obtained similar results considering higher number of particles in the ensembles as for instance 500,000 or 5,000,0000?. Additionally it would be useful if a date is included in each frame for movies S2, S3 and S4.

**Author's response:** To consider ensembles comprised of 5,000,000 particles and complete modelling within reasonable time, a rather massive (~1000 cores) computational cluster would be required, which was not available to me. However, I have conducted a sensitivity test using 500,000 particles for the random leeway factor model, location of the origin #23 (see Tab. 1). The results appear to be similar to those using 50,000 particles, as summarized in Table RC1. 1. I have also included respective animation into the Supplement.

In summary, I think the forward drift studies, which use ensembles of 50,000 particles are sufficiently accurate for the analysis. In comparison, Pattiaratchi and Wijeratne (2016) also used 50,000 particles per ensemble, while Jansen et al. (2016) – only 5,000.

**Table RC1. 1** Comparison of the computed percentages of particles washed ashore in various counties by the end of 2016 in dependence on number of simulated particles for the screened origin #23.

| Number of particles | Andaman Isl. | Australia | India | Indonesia | Kenya | Madagascar | Mauritius | Mozambique | La Reunion | Somalia | South Africa | Sri Lanka | Tanzania | Overall beached | Escaped domain |
|---|---|---|---|---|---|---|---|---|---|---|---|---|---|---|---|
| **50,000** | 0.078 | 0.854 | 0.022 | 0.048 | 0.300 | 12.166 | 1.966 | 3.068 | 2.338 | 0.272 | 1.946 | 0.090 | 0.952 | 25.430 | 0.724 |
| **500,000** | 0.090 | 0.866 | 0.037 | 0.037 | 0.329 | 12.163 | 1.976 | 3.097 | 2.438 | 0.289 | 1.908 | 0.087 | 0.977 | 25.703 | 0.744 |

With regard to the dates in animations S2, S3 and S4: they were indicated in each frame at the left top corner.

**Manuscript changes:** Table RC1.1 is added as Table 3 into the manuscript, and a respective paragraph in Section 3.3.
* * *
Comment #5. The random leeway factor used to produce figure 5, is a random distribution across the particle ensemble fixed in time or randomness varies in time?. Please clarify.

**Author's response:** Fixed in time.

**Changed to manuscript:** Figure 5 is devoted to results, so I think it is better to make this clearer in the description of the model. The statement "A random leeway factor assigned to a particle in ensemble was constant in time." is added in the text where figure 3 (distribution) is cited, as well as in the caption of Figure 3.
* * *
Comment #6. The paragraph explaining figures 5, and 6 and table 2 is unclear and thus the information contained in those figures and table is not understandable. The search areas marked in grey tone in Figure 5 were not considered during the whole period. For instance grey areas in the 1st column of figure 5 are early search areas, that were not considered in April. Why they appear in the second column? In the second column the grey areas appearing in the upper part were considered in later dates. Why are they sketched for April 5? Search areas change with time, therefore could the author explain how he defines cumulative coverages and why they say something in this case? Could the author define how the magnitude represented in the vertical axis of figure 6 (Maximum daily coverage) is defined? What is the meaning of the color code in the bars of figure 6? What case, random or constant leeway, is explained in the paragraph from line 47 to 57? To what figure refers that paragraph? Table 2 is unclear: what kind of leeway uses?; what is the meaning of the dates there?; how magnitudes appearing in the upper row 7are defined?.

**Author's response (by questions):**

**Comment #6.1**. The search areas marked in grey tone in Figure 5 were not considered during the whole period. For instance grey areas in the 1st column of figure 5 are early search areas, that were not considered in April. Why they appear in the second column? In the second column the grey areas appearing in the upper part were considered in later dates. Why are they sketched for April 5?

I followed AMSA style in the attempt to present 'cumulative' coverage maps, although I agree this approach might be somewhat confusing. The grayed areas represented all the areas searched up to certain specified date, e.g. in April 5 figure grayed areas corresponded to the areas surveyed before April 5. Showing a single day area would be less informative because during some intervals search could be conducted in likely irrelevant areas, in particular too far south during the first 10 days of the campaign. This seem to be difficult to reflect in a single figure, so I amended Fig. 5, and supplemented paper with animation S1 instead of Figure S1. This way of presentation does not affect the characteristics depicted in Table 2 and Figure 6.

**Comment #6.2**. "Search areas change with time, therefore could the author explain how he defines cumulative coverages and why they say something in this case?"

Yes, search areas changed in time, likewise particles in ensembles changed their locations. The cumulative percentages were defined as the sum of ensemble coverages during each day of the search operation. I think such a criteria could be important: for example, if a search campaign lasts for 10 days, what would result in a higher probability of debris detection: 100%-coverage of the debris field during only a single day, or 10% coverage during each day? In both the examples, the cumulative coverage defined in such a way would be 100%. In this case the overall probability to detect debris would depend on other factors, such as weather conditions.

**Comment #6.3**. "Could the author define how the magnitude represented in the vertical axis of figure 6 (Maximum daily coverage) is defined?"

These represent five largest daily coverages for each ensemble. Firstly, daily coverages were computed for each $j$-th ensemble for each $k$-th day of the aerial search campaign:

$$E_{j,k} = \left( \frac{1}{N} \sum_{\Omega_k} i_{j,k} \right) * 100\%,$$

where $N = 50,000$ (number of particles in an ensemble), $\Omega_k$ is the survey area on the $k$-th day (digitized from AMSA maps), $i_{j,k}$ are the particles of $j$-th ensemble, which were in $\Omega_k$ area on the $k$-th day.

After values of $E_{j,k}$ were obtained, they were sorted in descending order for each $j$-th ensemble (associated with $j$-th origin). The 5 largest of them are depicted in Fig. 6 for each ensemble in the form of bars of different colors.

**Comment #6.4**. "What is the meaning of the color code in the bars of figure 6? "

Black bar – the first largest (i.e. maximum),  red bar – the second largest, yellow bar – the third largest, green bar – the fourth largest, blue bar – the fifth largest.

**Comment #6.5**. "What case, random or constant leeway, is explained in the paragraph from line 47 to 57? To what figure refers that paragraph?"

I am not sure which lines are meant as these lines do not appear to be present in a single-column version of the manuscript; in double-column version concluding paragraphs are related to the random leeway model.

**Comment #6.6**. "Table 2 is unclear: what kind of leeway uses?; what is the meaning of the dates there?; how magnitudes appearing in the upper row 7are defined?."

Characteristics summarized in the Table 2 were for the random leeway factor model (it was stated in the caption of the Table 2).

'Date' column depicts date of the maximum coverage occurrence, that is to say when the maximum coverage depicted in the first column occurred.

The upper row of Table 2 depicts maximum coverages, their respective dates of occurrence, and cumulative coverages, for the origins #1, #11, #21, and #31 (the table's layout is arranged as 4 groups of columns and 10 rows to utilize space more efficiently). Respectfully, I am not sure what was meant under "magnitudes appearing in the upper row".

**Changed to manuscript:** I have amended Section 3.1 to clarify these questions, and revamped Fig. 5, so that hopefully revised manuscript became clearer. In particular, I added Eq. (19) to avoid ambiguities in the definition of the daily coverage. Also, instead of figure S1 in the Supplement, I included animation showing computed position of selected ensembles and daily coverage areas digitized from AMSA maps.
* * *
**Comment #7.** The autor states in lines 25-27 that: If the SST at particle's location never reached 23oC, or never dropped below 19oC after that, such a particle received zero score. However I do not see how this occurs by evaluating Eq. (18). For instance a particle which never reaches 23oC but drops to 16oC in the second period would score 0 in the first interval, 1 in the second time interval and 0 in the third one if it remains at 16oC. This according to (18) would score (0+1+0)/3 and this is not zero as stated by the author. Please clarify.

**Author's response:** Please note the difference between definition of the score $S_{i,\theta}$ and the auxiliary variable $\hat{S}_{i,\theta}$ in the formulations (18). In your example above $\hat{S}_{i,\theta} = \frac{0+1+0}{3}$, however the score received by the $i$-th particle $S_{i,\theta} = 0$ because there is no such time $t_1$, for which both the required conditions are met:

$$(a). \quad \max_{0 \leq t \leq t_1} \theta_i(t) \geq 23°C, \text{ and}$$

$$(b). \quad \min_{t_1 \leq t \leq t_2} \theta_i(t) \leq 19°C.$$

**Changes to manuscript:** I moved equation number (18) to $S_{i,\theta}$ and eliminated the auxiliary variable $\hat{S}_{i,\theta}$.
* * *
**Comment #8.** The information contained in figure S5 could be better interpreted if that figure is replaced by a movie made of the 40 frames displayed in the figure. It would be more intuitive if each frame included the initial point of the ensemble by marking it on the map instead of giving its geographical coordinates. The movie should include also the heading of the figure and the colorbar. The remaining information contained at the bottom of figure S5 could stay in a figure.

**Author's response:** One of the purposes of figure S5 is to allow for visual comparison of spatial distribution of washed ashore debris computed for different locations of the origins, which would be

more complicated if the figure is replaced with animation, in my opinion. I think animations are more suitable (intuitive) to demonstrate changes in time. In addition, I am not sure what the Referee #1 meant under "The remaining information contained at the bottom of figure S5 could stay in a figure": should the figure S5 stay, or should its bottom section to become a new figure, or should that static part of Figure S5 be included into an animation? Anyhow, I created animation per this request.

**Changes to appendix:** I have indicated origins in figure S5, and also created a respective animation S5. However, I would prefer Figure S5 instead of Animation S5 for the reason I stated above.
* * *
**Note other corrections**: As requested by Referee #2; the flaperon was discovered on July 29, not August 29 – corrected through the text of the manuscript; bars are added to indicate deterministic nature of the velocity components according to the commonly used convention; minor tweaks in the text.

**AUTHOR'S RESPONSE TO THE ANONYMOUS REFEREE'S #2 INTERACTIVE COMMENT.**

**General comments:** The disappearance of the Boeing 777-200ER of Malaysian Airlines in early 2014 is one of recent history's big mysteries. It is important for us as scientists and engineers to stay relevant to public interests and questions. I would like to see even more studies like the one presented here. I found the manuscript very interesting and relevant. The structure of the paper is well thought through and reads logically. I also found the figures very helpful in communicating both the study set-ups and finally the results. The use of biological information obtained from the barnacles on the recovered debris was probably my favourite part of the study especially when the modelling results could explain the temperature variations associated with the barnacle growth. In general, the four scenarios modelled, was also well constructed and adequately linked with the various debris drift scenarios.

**Author's response:** Thank you for your interest in my work, and for positive feedback.
* * *
**Specific comments.** The author did an adequate literature review. I would have like to see a bit more time spent on reviewing and/ or referencing other studies using a similar statistical approach. Or at least it would have been nice to get a clearer idea of how this drift study compares with other drift studies, not just studies related to MH370 debris. This is just a comment and not needed to make the manuscript acceptable. The coverage of other studies investigating the MH370 debris was well done and reads nicely. A non-technical person could read the introduction and get a clear idea of the context and relevance of the study. The three main aims listed on page 3 I also agree with. Refining the efficacy of search and rescue campaigns are crucial for future campaigns. As mentioned before, the cross correlation of the barnacle growth to surrounding water temperatures is my favourite part of the study and challenges us as scientist to think wider when it comes to answering scientific questions. I like the method of narrowing down the drift particles associated with debris discoveries on page 16. If the qualification 'window' for a particle was made to strict potential particle candidates, and thus traveling paths, might have been overlooked. As part of the conclusion it was reassuring to see that the study results agree with the high-priority search zone by the ATSB in June 2014.

**Author's response:** MH370 case is unique in terms of debris drift analysis. Firstly, this is because fragments are scattered across the whole Indian Ocean. Secondly, because currents and winds in these remote areas are relatively poorly studied. Thirdly, because recovered aircraft's fragments are mainly light-weight composite honeycomb structures having the shapes of thin plates, in contrast to ship containers, buoys, boats, etc., which were subjects of a number of previous studies. Also, I am not aware of other published works that would include temperature analysis to help in establishing drifting debris origin.

**Changed to manuscript:** I added 2 references in this regard: Daniel et al. (2002) and Breivik et al. (2011).
* * *
**Technical correction #1:** As I mentioned before I would have like to see more references.

**Author's response:** In attempt to balance between the references related to the drift studies, ocean modelling, turbulence, math, numerical methods, and MH370-specific papers, I added 2 new references related to the drift studies, which use similar approach.

**Changed to manuscript:** 2 new references are added: Daniel et al. (2002) and Breivik et al. (2011).

**Technical correction #2:** In general, it is nice to read the introduction to a figure before the figure appears in the manuscript. The same goes for tables.

**Author's response:** There is difference between placement of figures and tables is the single- (OSD) and double-column (OS) LaTex template. Originally I aimed to place figure or table on the same page where it was first referenced in a double-column layout. Should my manuscript be accepted, I will pay attention to this.
* * *
**Technical correction #3:** Equation 8 on page 6 was confusing due to unbalanced brackets. Please correct. Also, please reference literature used in this derivation or reference other studies using a similar technique or use literature to justify your approach.

**Author's response:** Thank you for pointing this out. Brackets are corrected in revised manuscript – please note it became Eq. (10). As long as the Referee #1 also requested for the clarifications related to this equation, I have revamped the whole Section 2.1.3. In particular, references to Daniel et al. (2002) and Breivik et al. (2011) were included, along with justifications relevant to horizontally floating thin plates.

**Changed to manuscript:** Revamped Section 2.1.3.
* * *
**Technical correction #4:** Equation 10 on page 7 the drag coefficient CDs must surely be CDw?

**Author's comment:** Yes, thank you for pointing this out.

**Changed to manuscript:** Corrected.
* * *
**Note other changes**: As requested by Referee #1; the flaperon was discovered on July 29, not August 29 – corrected through the text of the manuscript; bars are added to indicate deterministic nature of the velocity components according to the commonly used convention; minor tweaks in the text.

---

## Author Response (AR2)

**Topic Editor Decision: Publish subject to technical corrections** (25 Apr 2018) by John M. Huthnance
Comments to the Author:
Thank-you for your revised manuscript addressing the referees' comments. I am now asking you to consider the "Technical Corrections" below after which it will go directly to the Copernicus publication process. This will involve copy editing which I expect will make changes to the use of English. Please therefore check that the final version keeps your intended meaning.
Thank-you for submitting to Ocean Science.

Author's response: Thank you for your decision, and thank you for your help in improving my manuscript. Detailed responses are in blue.

Page and line numbers numbers are as in the 2-column version that I now have.

Page 1.
Line 50. Meaning of "respective region" unclear. Delete "has".

Modified into "…*whose geographical responsibility for rescue and recovery covers a region of the Indian Ocean where the terminus of 9M-MRO path could have been located according to the satellite data*…"

Line 55. ". . April 27, 2014," ? Added.

Page 2.
Line 35. "what" -> "which". Corrected.

Line 81. "walking" -> "walk". Corrected.
Line 94. "which could have reached . . by July 29, 2015 and could have been subjected . ." Corrected.

Page 3.
Line 15. ". . other hand, the large study domain . ." Corrected.
Line 29. ". . ellipsoid, as follows" Corrected.

Page 5.
Line 27. Better ". . in one of the four ensembles of simulations (section 2.2)." ? [To reinforce the relation of this section to what follows.] Modified as requested.
Lines 45 and 52 imply $\sigma**2 = 1/2$. I might not have followed the statistics correctly but this seems to be a choice affecting the distribution in figure 3, which looks to have most leeway factors less than those mentioned in section 2.1.3, and likewise mostly less than the 3.29% or even the 2.76% of the other ensembles in section 2.2. Does this give the random leeway factor ensemble a bias towards wind-forced motion? Information about the sensitivity of the results to this is of course available in the results of the four ensembles of section 2.2. Maybe a short comment about this could be added, at the end of section 3 or in section 4?

In brief, this choice of σ does not give the random leeway factor ensemble a bias towards wind-forced motion, although it affects distribution shown in Fig3. The mean leeway factor of the particles of the proposed distribution is approximately 2%, which is lower than the windage of 3.3% I would expect for a thin horizontally floating object (which is consistent with the experimental data for the flaperon established by DGA, although CSIRO argues that it is lower). This decrease of the leeway factor is mainly due to partial submerging rather than the choice of σ. In other words, the random model biased against wind-induced motion compared to thin horizontally-floating objects. In comparison, Jansen et al. (2015) studied 0.5% to 2.5% with the discrete increments of 0.5%; Griffin et al. (2015-2017): 0.0%, 1.0%, 1.2%, 1.8%, 2.8%, and 3%.

Page 6.
Line 14. ". . in a variety . ." Corrected.

Line 17. ". . displacements exceeding . ." Corrected.
Line 57. "Close to a shore . ."? [delete "the" in any case] Corrected.
Line 65. Delete "of". Corrected.
Line 72. "arithmetic." Corrected.

Page 7.
Line 2. "scenarios". This should be the same word as later where "model" is mostly used (e.g. lines 12, 14; also in figure 5). I am not sure that "model" is the best word but is probably the most practical. Then better (for line 2) "Four scenarios – hereafter "models" – with respect to . .").
Line 19. "The study domain extended from 20°E . ." Modified as requested.
Line 28. "The following datasets were used in this study . ." Modified as requested.

Page 8.
Figure 5 caption. ". . originating from . . for two models . ." Corrected.
Line 29. ". . from nine . ." Corrected.
Line 39. Delete "of" Corrected.

Page 9.
Line 11. "what" -> "which". Corrected.
Line 22. ". . originating from . ." Corrected.
Line 25. ". . originating from" Corrected.

Page 11.
Line 55. ". . as of April 2017 . ." [meaning any time up to April; not only in April] Corrected.
Line 58. "at Thirsty Point. . ." Corrected.

Page 13.
Line 1. "An example of the so-estimated concentration . ." Modified as requested.
Line 11. ". . concentration near Cape Leeuwin" Modified as requested.
Line 18. ". . landed near Thirsty Point . ." Modified as requested.
Line 28. ". . object, it had covered a long distance . ." Corrected.

Page 14.
Lines 35-37. Better order might be "Considerable beaching, particularly in Australia and Sri Lanka where no fragments were found; . ." However this does give a slightly different emphasis which you may not want. Otherwise, please remove "," after "beaching".

"," removed as the former version changes intended meaning.
Lines 40-42. Similar (but not quite the same); please remove "," after "countries". Removed.
Line 44. ". . originating from . ." Corrected.
Line 51. "to Thirsty Point." Corrected.
Line 62. "declares" Corrected.
Line 63. "Acknowledgements. Use . ." Modified as requested.

Other changes:

Page 1.

Line 57. "on March 18" is deleted and replaced with "…from the air…" [debris were spotted not only on a single day].

Page 2.

Line 39. "…study conducted…" replaced with "…report published…" [too many repetitions of 'study'].

Line 52. "… The latest CSIRO study…" removed to make it shorter, i.e. "Griffin et all (2017) disagrees…"

Line 101. "model" replaced with "study".

Page 3.

Line 13. Added "On the one hand…"

Line 17, 22. "Transformation…" is moved into the beginning of sentence.

Page 4.

Line 26. "… as applied in Daniel et al. (2002) and Breibik et al. (2011) to study drift of ship containers"

Line 27. "horizontally" changed into "nearly horizontally"

Line 33. "dynamic velocity" changed into "friction velocity".

Line 51. "for ship containers" removed.

Line 64. "x 0.25 m" added.

Line 69. "typically" replaced with "normally"

Line 70. "reduced to" replaced with "approximated by"

Page 5.

Line 15. "In those scenarios" is replaced with "In two of the four considered models" [to make it in line with the requested changes].

Line 30. "a slightly tilted orientation" replaced with "slightly tilted orientations".

Line 43. "0-" changed into "0 – " [space added].

Line 46. "," removed.

Page 7.

Line 17. Added "constant in time" and "all the 40 ensembles (Section 2.2.1) were identical in terms of the particles they were comprised of" [to stress that the leeway factors did not change in time].

Page 8.

Line 15. *"…to understand contribution of wind to these errors (e.g., Griffin et al. (2017) suggests that the average leeway factor of the GDP buoys is around 1.2%*), … " added.

Page 9.

Line 29. "goals" changed into "major goals"

Page 10.

Fig. 7 caption: "LW: leeway factor; DA: drift angle" added [abbreviation in figures].

Page 11.

Line 21. "are shown in Figure 7" moved to the end of sentence.

Line 23. "fitting" replaced with "fittings".

Line 31. "predicted" changed to "indicated".

Line 36. "corresponding" inserted, "'released' there" removed.

Line 43. "," added.

Line 67. "S2" replaced with "S2a" [there are 2 animations in the supplement corresponding to this simulation, one using 50,000 particles, and the other one – using 500,000].

Page 12.

Figure 9. "moonsoon" corrected to "monsoon" [typo error].

Page 13.

Line 4. "Figure and Animation S5".

Page 14.

Line 10. "Respective animation S2b is included into the Supplement" added.

Line 21. "with regard to" changed to "with respect to".

Also units in axis captions are indicated with parentheses to make style uniform though the manuscript.